# USP52 acts as a deubiquitinase and promotes histone chaperone ASF1A stabilization

Shangda Yang[1], Ling Liu[1], Cheng Cao[1], Nan Song[1], Yuejiao Wang[1], Shuai Ma[1], Qi Zhang[1], Na Yu[1], Xiang Ding[2], Fuquan Yang[2], Shanshan Tian[1], Kai Zhang[1], Tao Sun[3], Jie Yang[1,4], Zhi Yao[4], Shaoyuan Wu[1] & Lei Shi[1,4]

Histone chaperone ASF1A has been reported to be dysregulated in multiple tumors; however, the underlying molecular mechanism that how the abundance and function of ASF1A are regulated remains unclear. Here we report that ASF1A is physically associated with USP52, which is previously identified as a pseudo-deubiquitinase. Interestingly, we demonstrate that USP52 is a bona fide ubiquitin-specific protease, and USP52 promotes ASF1A deubiquitination and stabilization. USP52-promoted ASF1A stabilization facilitates chromatin assembly and favors cell cycle progression. Additionally, we find that USP52 is overexpressed in breast carcinomas, and its level of expression correlates with that of ASF1A. Moreover, we reveal that impairment of USP52-promoted ASF1A stabilization results in growth arrest of breast cancer cells and sensitizes these cells to DNA damage. Our experiments identify USP52 as a truly protein deubiquitinase, uncover a molecular mechanism of USP52 in chromatin assembly, and reveal a potential role of USP52 in breast carcinogenesis.

[1] 2011 Collaborative Innovation Center of Tianjin for Medical Epigenetics, Tianjin Key Laboratory of Medical Epigenetics, Key Laboratory of Breast Cancer Prevention and Therapy (Ministry of Education), Department of Biochemistry and Molecular Biology, Tianjin Medical University, Tianjin 300070, China. [2] Laboratory of Proteomics, Institute of Biophysics, Chinese Academy of Sciences, Beijing 100101, China. [3] State Key Laboratory of Medical Chemical Biology and College of Pharmacy Nankai University Tianjin 300071, China. [4] Tianjin Key Laboratory of Cellular and Molecular Immunology, Key Laboratory of Immune Microenvironment and Disease (Ministry of Education), Department of Immunology, School of Basic Medical Sciences, Tianjin Medical University, Tianjin 300070, China. These authors contributed equally: Shangda Yang, Ling Liu. Correspondence and requests for materials should be addressed to L.S. (email: shilei@tmu.edu.cn)

Histone chaperones play critical roles at all stages of DNA transactions[1–5]. In general, chaperones accompany with histones upon their synthesis, escort them into the nucleus, and facilitate their specific association or dissociation with chromatinized DNA[6–8]. Certain histone chaperones have been assigned to promote specific nucleosome assembly pathways, a critical step towards chromatin restoration on newly synthesized or repaired DNA[3,9–16]. Appropriate deposition of histones by dedicated escorting machinery is important in shaping the chromatin landscape thus cellular homeostasis, while failure to do this is associated with distinct diseases including cancers[17,18].

The histone H3–H4 chaperone anti-silencing function 1 (ASF1) regulates chromatin structure organization, through delivering canonical S-phase histones H3.1–H4 to chromatin assembly factor 1 (CAF1) in a replication-coupled chromatin assembly process as well as transferring variant histones H3.3–H4 to histone regulator A (HIRA) or DAXX/ATRX complex in a DNA synthesis-independent manner[15,17,19–24]. Additionally, ASF1 cooperates with the MCM2-7 replicative helicase to regulate histone recycling in replication fork progression, through handling histones from the front of the replication forks onto newly synthesized DNA strands[16,25]. Mammalian cells have two ASF1 homologs, ASF1A and ASF1B, with largely redundant roles in histone eviction and deposition[26,27]. Recent studies indicate that histone chaperone ASF1A, but not ASF1B, in mammals, facilitates acetylation of histone H3 lysine 56 (H3K56Ac), an important histone mark in packaging DNA into chromatin following DNA replication and repair in eukaryotic cells[18,28,29]. Interestingly, the expression of ASF1A and the level of H3K56Ac are elevated in multiple types of tumors and positively correlate with each other[18], suggesting that aberrant regulation of ASF1A-deposited H3K56Ac is associated with tumor progression. Given the role of histone depositions in maintenance of higher order chromatin structures, in particular, genome stability and epigenome inheritance, histone supply pathways must be fine-tuned[1,2,8]. Therefore, understanding how the abundance of ASF1A is regulated in physiological state and how it is dysregulated in malignancies is of great importance to the understanding of genome/epigenome integrity and tumor development, respectively.

The protein homeostasis in cells is largely governed by the ubiquitin–proteasome system[30–33]. This system is involved in multiple cellular activities including cell growth, apoptosis, and death, while its dysregulation is associated with various pathological disorders, including malignancy[32,34–36]. Ubiquitin conjugation is mediated via an E1–E2–E3 cascade, while ubiquitin removal is catalyzed by deubiquitinating enzymes (DUBs), a group of proteins comprising approximately 80 active members in mammals[31,37]. The ubiquitin-specific peptidase 52/poly(A) nuclease 2 (USP52/PAN2), a member of the ubiquitin-specific protease (USP) superfamily, contains a WD40 repeat domain at the N terminus, a ubiquitin C-terminal hydrolase (UCH) domain, and a C-terminal RNase domain of the DEDD superfamily[31,38]. USP52 has been well characterized as a poly(A) nuclease in the PAN2/PAN3 deadenylation complex[39,40], and a recent study reported that USP52 is a key component of P-body (processing body) and functions to prevent *HIF1A* mRNA degradation[38]. Yet, whether USP52 is capable of removing ubiquitin linkages remains an open question, although crystal structure analysis of its yeast or fungi orthologue indicated that the UCH domain lacks catalytic residues required for protease activity and is incompatible with catalysis[41,42].

In this study, we report that USP52 is able to remove ubiquitins either from specific types of polyubiquitin chains or K48-linked polyubiquitinated ASF1A. We reveal that USP52 promotes chromatin assembly through stabilizing ASF1A, and point a role of USP52 in breast carcinogenesis and cellular resistance of breast cancer cells to DNA damage.

## Results

**Histone chaperone ASF1A is physically associated with USP52.**
In an effort to better understand the mechanistic role of ASF1A in chromatin assembly and tumorigenesis, we employed affinity purification and mass spectrometry to identify ASF1A-associated proteins. The results indicate that ASF1A was copurified with a number of proteins, including CAF1, NASP, NPM, DAXX, Codanin, TLK1, TLK2, and RIF1, most of which are known to interact with ASF1A[3,26]. Interestingly, USP52, a member of protein deubiquitinases carrying a UCH domain, was also identified in the ASF1A-containing protein complex (Fig. 1a and Supplementary Data 1). To confirm the in vivo association of ASF1A with USP52, co-immunoprecipitation experiments were performed and the results showed that USP52 was efficiently co-immunoprecipitated with ASF1A, and vice versa (Fig. 1b). However, co-immunoprecipitation analysis indicated that PAN3 was physically associated with USP52 as reported[39,40], but not with ASF1A (Fig. 1b). Interestingly, ASF1A, but not USP52, could be co-immunoprecipitated with FLAG-tagged Histone H3.1 or Histone H3.3 (Supplementary Fig. 1a). Although ASF1B shares high similarity with ASF1A in primary sequences, USP52 could not be co-immunoprecipitated with ASF1B (Supplementary Fig. 1b). These results suggest that ASF1A and USP52 co-exist in a protein complex without PAN3 and Histone H3.

To gain further support of the in vivo interaction between ASF1A and USP52, protein fractionation experiments with FLAG-ASF1A affinity eluates were carried out. The results indicate that the majority of the purified FLAG-ASF1A existed in a multiprotein complex, which peaked in fractions from 9 to 15 with USP52 (Fig. 1c). Moreover, co-immunoprecipitation analysis with protein fractionations from different cellular compartments indicated that USP52 could be co-immunoprecipitated with cytosolic, but not nuclear, ASF1A (Fig. 1d), and immunostaining followed by confocal microscopy analysis confirmed that USP52 mainly co-localizes with ASF1A in cytoplasm (Fig. 1e). These results support the observation that ASF1A is physically associated with USP52 in vivo and suggest that ASF1A interacts with USP52 in cytoplasm.

Next, co-immunoprecipitation analysis indicated that the N-terminal WD40 repeat domain of USP52 (USP52/WD40) is responsible for the association of USP52 with ASF1A (Fig. 1f), and the first 30 amino acids in the N terminus of ASF1A are sufficient and necessary for the interaction of ASF1A with USP52 (Fig. 1g). Furthermore, in vitro pull-down experiments demonstrated that ASF1A is capable of interacting with full-length USP52 and USP52/WD40 (Supplementary Fig. 1c), and USP52/WD40 directly interacts with full length and the first 30 amino acids in the N terminus of ASF1A (Supplementary Fig. 1d). Collectively, these results indicate that ASF1A interacts with USP52 through the N-terminal region of ASF1A and the WD40 repeat domain of USP52.

**USP52 is a bona fide deubiquitinase.** Since PAN3 is absent from the USP52/ASF1A protein complex, we deducted that the effect of USP52 on ASF1A functionality is independent of its deadenylase activity. Primary structure alignment indicated that *Homo sapiens* USP52 lacks a commonly shared catalytic triad (cysteine, histidine, and aspartate) among USPs[43] (Supplementary Fig. 2a) and crystal structure analysis of USP52 from *Neurospora crassa* or *Saccharomyces cerevisiae* pointed out its UCH domain is incompatible with catalysis[41,42]. However, given that specific

protein folding or post-translational modifications could impact on higher structure conformation and potentially coordinate or rearrange the catalytic center of enzymes[44], it will be interesting to investigate whether USP52 is capable of biochemically hydrolyzing ubiquitin linkages. To test this hypothesis, in vitro deubiquitination assays were performed with K48-linked ubiquitin linkages and bacterially expressed or insect cells expressed UCH domain of USP52 (USP52/UCH). The results showed that the bacteria-purified USP52/UCH has no catalytic activity as

previously reported[43], even the incubation time was prolonged to 8 h (Fig. 2a), while the insect cells-purified recombinant was able to cleave the K48-linked ubiquitin conjugates with an accumulation of low molecular weight of ubiquitin conjugates in a time-dependent manner and the evident cleavage effect was observed post 1 h of incubation (Fig. 2b). Since post-translational modifications or higher order structures modulate protein function in most eukaryotes, we speculate that the UCH domain of USP52 purified from Sf9 cells likely undergoes conformational changes

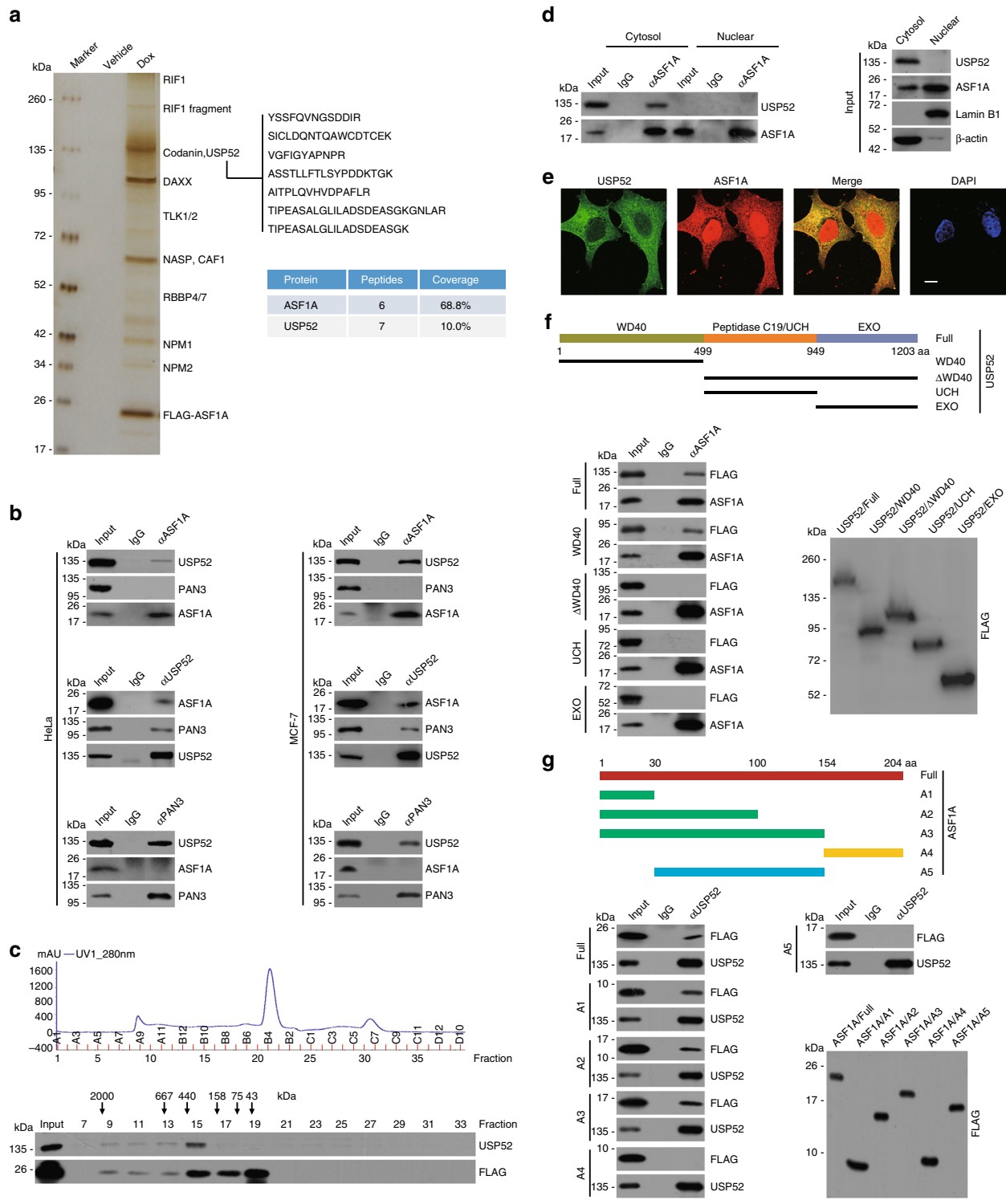

to gradually accommodate with ubiquitin substrates thus eventually elicits detectable enzymatic activity, albeit with a slower kinetics than canonical DUBs that usually starts hydrolyzing tri- or tetra-ubiquitin linkages after minutes of reaction and almost fully cleaves the isopeptide bond in less than 1 h[45,46].

To differentiate the selectivity of USP52 against distinct types of ubiquitin chains, we next performed in vitro deubiquitination assays with increasing amounts of recombinant USP52/UCH and K6-, K11-, K27-, K29-, K33-, K48-, K63-, or M1-linked tetra- or di-ubiquitin linkages. The results indicated that, although bacterially expressed USP52/UCH exhibited non-detectable catalytic activity (Supplementary Fig. 2b), K6-, K11-, K48-, K63-, and M1-linked ubiquitin chains could be hydrolyzed by from Sf9 cells-purified USP52/UCH in a dose-dependent manner (Fig. 2c). To further substantiate the deubiquitinase activity of USP52, we purified full length or UCH domain deficient USP52 (USP52/ΔUCH) from HeLa cells with high salt and detergent containing buffer. In vitro deubiquitination assays demonstrated that wild-type USP52, but not USP52/ΔUCH, was able to trim K6-, K11-, K48-, K63-, and M1-linked ubiquitin chains (Fig. 2d). Together, these results support the argument that USP52 is capable of cleaving specific types of ubiquitin linkages and it is a bona fide deubiquitinase.

Although the conserved cysteine residue in the triad of classical DUBs is replaced by alanine (A526) in the UCH domain of USP52 (Supplementary Fig. 2a), we demonstrated that addition of the alkylating reagent N-ethylmaleimide (NEM), a known inhibitor of cysteine proteases[47,48], almost completely abolished the catalytic activity of USP52 against K48- or K63-linked tetra-ubiquitins (Fig. 2e). In these experiments, USP7 was taken as a positive control (Fig. 2e). The sensitivity of the UCH domain of USP52 to NEM suggested the involvement of one or more cysteine residues in catalysis. We hypothesized that C528 or C530, adjacent to A526, likely rearranges to form newly active center to hydrolyze ubiquitin chains. However, in vitro deubiquitination assays revealed that C528A/C530A mutant is still active in hydrolyzing K48- or K63-linked tetra-ubiquitins, albeit with an evidently lower efficiency (Fig. 2f), indicating that one or both of the two cysteine residues is important but not essential for catalysis. Although structure or more mutational analysis is needed to further understand the underlying mechanism of USP52 catalytic activity, these results support the notion that USP52 possesses USP activity.

**USP52 deubiquitinates ASF1A.** Since ASF1A is reported as a ubiquitinated protein[49], we then asked whether USP52 functions to deubiquitinate ASF1A. In vitro deubiquitination assays with HA-Ub-conjugated FLAG-ASF1A and Myc-USP52 purified from HeLa cells by high salt and detergent containing buffer revealed that USP52 was capable of deubiquitinating ASF1A in a dose-dependent manner (Fig. 3a). Next, we demonstrated that the levels of ubiquitinated FLAG-ASF1A species dramatically increased in USP52-depleted cells (Fig. 3b) and decreased in USP52 highly expressing cells (Fig. 3c). Meanwhile, we assessed the ubiquitination of endogenous ASF1A, and the results confirmed that endogenous ASF1A is a ubiquitinated protein and showed that USP52 overexpression resulted in a decrease in the level of ubiquitinated ASF1A species (Fig. 3d). Corroborating with the finding that WD40 repeat domain is required for the physical association of USP52 with ASF1A, expression of USP52 lacking WD40 repeat domain (USP52/ΔWD40) did not affect the level of ubiquitinated ASF1A species (Supplementary Fig. 3a). Meanwhile, USP52/ΔUCH had no catalytic activity against poly-ubiquitins conjugated on ASF1A, although USP52/ΔUCH could be efficiently precipitated by FLAG-ASF1A (Supplementary Fig. 3a).

Then, we utilized ubiquitin mutant with all lysine residues replaced by arginine except K48 (K48-only) or K63 (K63-only) to differentiate ubiquitin moieties of polyubiquitinated ASF1A cleaved by USP52. In vivo deubiquitination assays indicated that K48-linked ubiquitin conjugates are the major forms of ubiquitin linkages hydrolyzed by USP52 (Fig. 3e and Supplementary Fig. 3b). Consistently, in vitro deubiquitination assays further revealed that USP52 was capable of removing K48-linked ubiquitin chains on ASF1A (Fig. 3f), and NEM treatment almost abolished this effect (Supplementary Fig. 3c).

In order to characterize the ubiquitin conjugation sites of ASF1A that could be opposed by USP52, six lysine residues in ASF1A were individually replaced with arginine. In vivo deubiquitination assays showed an evident decrease in the polyubiquitination level of all ASF1A mutants except for K129R (Fig. 3g and Supplementary Fig. 3d), while the association of USP52 with all ASF1A mutants was comparable to that of USP52 with wild-type ASF1A (Supplementary Fig. 3e). Importantly, mass spectrometry analysis of ubiquitin sites on ASF1A confirmed that K129 carries di-glycine remnant after trypsin digestion (Fig. 3h and Supplementary Data 2). Next, in vitro deubiquitination assays revealed that USP52 was able to efficiently remove K48-linked polyubiquitin linkages of wild-type ASF1A (ASF1A/wt) and ASF1A/K41R, but not that of ASF1A/K129R (Supplementary Fig. 3f), and K48-linked polyubiquitin chains conjugated onto ASF1A/K129R was dramatically reduced comparing with that onto ASF1A/wt. Taken together, these data

**Fig. 1** Histone chaperone ASF1A is physically associated with USP52. **a** Immunoaffinity purification and mass spectrometry analysis of ASF1A-containing protein complexes. MCF-7 cells allow doxycycline (Dox)-inducible expression of stably integrated FLAG-ASF1A was generated. Whole-cell extracts from MCF-7 cells with or without FLAG-ASF1A expression were prepared and subjected to affinity purification using an anti-FLAG affinity column. After extensive washing, the bound proteins were eluted with excess FLAG peptides, resolved, and then visualized by silver staining on SDS-PAGE. The protein bands on the gel were recovered by trypsinization and analyzed by mass spectrometry. Detailed results from the mass spectrometric analysis are provided as Supplementary Data 1. **b** Whole-cell lysates from HeLa or MCF-7 cells were immunoprecipitated (IP) followed by immunoblotting (IB) with antibodies against the indicated proteins. **c** ASF1A-containing protein complex purified from HeLa cells stably expressing FLAG-ASF1A was fractionated by fast protein liquid chromatography (FPLC) on Superose 6 size exclusion columns with high salt buffer. Chromatographic elution profiles and IB analysis of the chromatographic fractions with antibodies against the indicated proteins are shown. The elution positions of calibration proteins with known molecular masses are indicated, and an equal volume from each fraction was analyzed. **d** Co-immunoprecipitation analysis of the interaction between USP52 and ASF1A with cellular lysates from different cellular compartments of MCF-7 cells. **e** Immunostaining and confocal microscopy analysis of USP52 and ASF1A subcellular localization in MCF-7 cells. Scale bar, 10 μm. **f** Co-immunoprecipitation analysis of the molecular interface between ASF1A and USP52 with cellular lysates from HeLa cells expressing FLAG-GFP tagged full length or deletions of USP52. The conserved domain of USP52 was determined by the SMART program. WD40: WD40 repeat domain, UCH: ubiquitin C-terminal hydrolase domain, EXO: exonuclease domain. **g** Co-immunoprecipitation analysis of the molecular interface between ASF1A and USP52 with cellular lysates from HeLa cells expressing FLAG-tagged full length or deletions of ASF1A

indicate that USP52 targets ASF1A/K129 for deubiquitination and USP52 is a bona fide deubiquitinase for ASF1A.

**USP52 stabilizes ASF1A.** Since K48-linked polyubiquitination is the canonical signal that marks proteins for degradation by

proteasome[50], we next examined whether USP52 controls the protein abundance of ASF1A. Western blotting analysis revealed that the level of ASF1A, but not PAN3 or DAXX, was significantly reduced upon USP52 depletion in MCF-7 cells (Fig. 4a, left panel). However, USP52 knockdown had minimal effect on

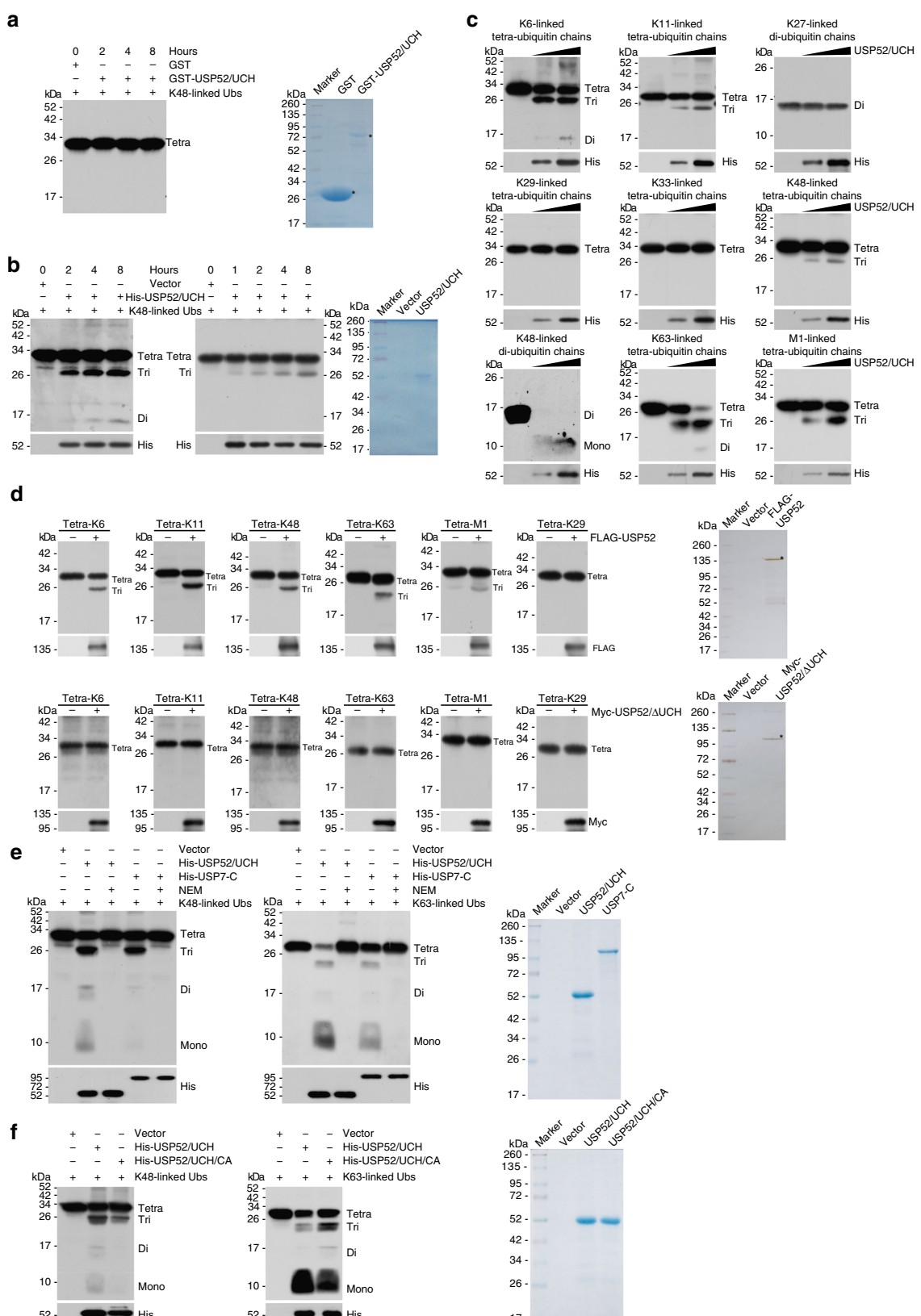

ASF1A mRNA level (Fig. 4a, right panel). Similar results were obtained in HeLa cells (Fig. 4b). Although knockdown of USP7, another deubiquitinase, resulted in downregulation of MDM2 as reported[51], it led to upregulation of ASF1A (Fig. 4c). These results suggest that ASF1A is specifically stabilized by USP52. Next, we found overexpression of USP52 was able to restore the expression of ASF1A in USP52-deficient cells (Fig. 4d). Meanwhile, over-expression of wild-type USP52, but not USP52/ΔWD40 or USP52/ΔUCH, resulted in an elevated expression of ASF1A (Fig. 4e). Moreover, the reduction in ASF1A protein level associated with USP52 depletion was probably through a proteasome-mediated protein degradation mechanism, as the effect could be effectively blocked by a proteasome specific inhibitor, MG132 (Fig. 4f). These observations suggest that ASF1A is a substrate of USP52.

Further supporting this deduction, cycloheximide (CHX) chase assays revealed that USP52 depletion was clearly associated with a decreased half-life of ASF1A (Fig. 4g and Supplementary Fig. 4a), while USP52 gain of function was associated with an increased half-life of ASF1A (Fig. 4g and Supplementary Fig. 4b). Moreover, depletion of USP52 was associated with a decrease in the half-life of ASF1A/wt and ASF1A/K41R, but not that of ASF1A/K129R (Supplementary Fig. 4c), suggesting that K129 is the major ubiquitination site targeted by USP52 for ASF1A stabilization. Then, we demonstrated that the protein levels of USP52 and ASF1A oscillated in a parallel pace during cell cycle progression: both increased in S and early $G_2$ phases and decreased in $G_1$ and $G_2$–M phases (Fig. 4h). Consistently, the physical association of USP52 with ASF1A was detected primarily in the S phase of the cell cycle (Fig. 4i). In addition, the expression levels of USP52 and ASF1A in multiple cell lines showed a positively correlated pattern (Fig. 4j). Together, these results strongly support the notion that USP52 controls the stability of ASF1A.

**USP52 promotes chromatin assembly through stabilizing ASF1A.** A remarkable role of ASF1A is that it presents newly synthesized histones H3–H4 to acetyltransferase complex for acetylation of histone H3K56 (H3K56Ac) and then hands them off to histone chaperone CAF1, which eventually deposits $(H3–H4)_2$ tetramers onto replicating DNA[18,52–54]. Chromatin assembly is routinely examined by MNase (micrococcal nuclease) cleavage assay, in which chromatins with high density of nucleosome occupancy are reluctant to be digested at low concentration of MNase and a reduction or an impairment of nucleosome occupancy results in chromatins more sensitive to MNase treatment[55–57]. Thereby, MNase digestion experiments were used to examine whether USP52 plays a role in replication-coupled chromatin assembly. Specifically, cells enriched in the S phase were harvested (Supplementary Fig. 5a), and chromatins were isolated. MNase digestion of chromatins from control cells displayed characteristic ladders indicative of mono-, di-, tri-, and multi-nucleosomes units, indicating that oligonucleosomes were formed onto newly replicated DNA (Fig. 5a, panel 1), while chromatins from cells with either USP52- or ASF1A-depleted showed an increased MNase digestion sensitivity, suggesting that histone deposition onto newly replicated DNA was impaired (Fig. 5a, panel 1). Meanwhile, chromatins from cells with forced expression of USP52 or ASF1A displayed similar degree of resistance to MNase cleavage (Supplementary Fig. 5b). These results indicate that USP52 is involved in replication-coupled nucleosome assembly.

Next, we showed that the defects of replication-coupled nucleosome assembly associated with USP52 depletion were ameliorated by Dox-inducible ASF1A expression (Fig. 5a, panel 2). Furthermore, MNase digestion resistance resulted by USP52 gain of function could be reverted by ASF1A depletion (Fig. 5a, panel 3). These results suggest that USP52 promotes chromatin assembly in an ASF1A-dependent manner. In favor of this argument, synergistically depletion of USP52 and ASF1A had no additive effect (Fig. 5a, panel 4). Additionally, we demonstrated that USP52/ΔWD40 or USP52/ΔUCH failed to promote replication-coupled nucleosome assembly (Fig. 5a, panels 5 and 6), further corroborating the notion that the functionality of USP52 in chromatin assembly is through interacting with and stabilizing ASF1A.

To corroborate the role of USP52-promoted ASF1A stabilization in chromatin replication, cellular extracts from cells used in MNase digestion assays were collected and analyzed by western blotting. In parallel with the alterations of chromatin assembly, the level of H3K56Ac was reduced upon USP52 depletion (Fig. 5b, panel 1), and elevated after forced expression of USP52 (Supplementary Fig. 5c), faithfully mimicking the effect of ASF1A interventions (Fig. 5b, panel 1 and Supplementary Fig. 5c). Moreover, gain or loss of function of ASF1A could largely restore the changes of H3K56Ac level in USP52-deficient or USP52-proficient cells, respectively (Fig. 5b, panels 2 and 3). Importantly, no additive effect was observed upon simultaneously depletion of USP52 and ASF1A (Fig. 5b, panel 4). Unlike wild-type USP52, overexpression of USP52/ΔWD40 or USP52/ΔUCH had no impact on the level of H3K56Ac (Fig. 5b, panels 5 and 6). These results favor the argument that USP52-promoted ASF1A stabilization regulates DNA synthesis-coupled chromatin assembly through ASF1A-mediated deposition of H3K56Ac.

To directly probe DNA synthesis, we labeled newly synthesized DNA with a short pulse of EdU (5-ethynyl-2-deoxyuridine)

**Fig. 2** USP52 is a deubiquitinase. **a** In vitro deubiquitination assays with *E. coli* cells-purified UCH domain of USP52 (USP52/UCH, 1 μg) and K48-linked tetra-ubiquitin linkages (1 μg) under different time points as indicated. The cleavage effect was examined by western blotting with antibody against ubiquitin. The asterisk indicates the recombinant protein stained by Commassie Blue. **b** In vitro deubiquitination assays with Sf9 cells-purified USP52/UCH (1 μg) and K48-linked tetra-ubiquitin linkages (1 μg) under different time points as indicated. The cleavage effect was examined by western blotting with antibody against ubiquitin. The asterisk indicates the recombinant protein stained by Commassie Blue. **c** In vitro deubiquitination assays with different types of ubiquitin linkages and increasing amounts of Sf9 cells-purified USP52/UCH (1 and 3 μg) or control eluents. After 4 h of incubation, the cleavage effect was examined by western blotting with antibody against ubiquitin. **d** In vitro deubiquitination assays with different types of ubiquitin linkages (1 μg) and HeLa cells-purified full-length USP52 (1 μg) or UCH domain deficient USP52 (USP52/ΔUCH, 1 μg) with high salt and detergent buffer. After 4 h of incubation, the cleavage effect was examined by western blotting with antibody against ubiquitin. The asterisk indicates the recombinant protein stained by silver staining. **e** In vitro deubiquitination assays with K48- or K63-linked tetra-ubiquitins (0.5 μg) and Sf9 cells-purified USP52/UCH (3 μg) in the presence or absence of 2 mM alkylating reagent *N*-ethylmaleimide (NEM). Sf9 cells-purified recombinant C-terminal USP7 (UCH domain and C-terminal ubiquitin-like domain, 1.5 μg) was taken as a positive control. After 4 h of incubation, the cleavage effect was examined by western blotting with antibody against ubiquitin. **f** In vitro deubiquitination assays with K48- or K63-linked tetra-ubiquitins (0.5 μg) and Sf9 cells-purified USP52/UCH (3 μg) or cysteine mutant of UCH domain (C528A/C530A, 3 μg). After 4 h of incubation, the cleavage effect was examined by western blotting with antibody against ubiquitin. CA represents mutant carrying both C528A and C530A (C528A/C530A)

incorporation, which can be visualized with Cell-Light technology. Like ASF1A deficiency, the proportion of cells with EdU staining was increased, whereas the amount of EdU incorporation was significantly reduced upon USP52 depletion, indicating an impaired DNA synthesis in S-phase cells (Fig. 5c and Supplementary Fig. 5d). Consistently, the alterations of PCNA (proliferating cell nuclear antigen), which normally acts as a processivity clamp for DNA polymerases thus marks sites of ongoing replication[58], are similar to that of EdU stainings (Fig. 5c), probably reflecting that less replication forks are active in USP52- or ASF1A-depleted cells. Importantly, we revealed that changes of EdU staining profile associated with USP52 loss or

gain of function could be largely reverted by ASF1A over-expression or knockdown, respectively (Fig. 5d). In favor of these observations, FACS analysis revealed that, similar to ASF1A knockdown, USP52 depletion also led to an accumulation of cells in the S phase, the effect of which could be offset by forced expression of ASF1A (Fig. 5e). Taken together, these results support the argument that USP52-promoted ASF1A stabilization is required for efficient DNA replication and proper S-phase progression.

**USP52/ASF1A signaling is implicated in breast carcinogenesis**. Dysregulation of ASF1A or/and H3K56Ac have been reported to

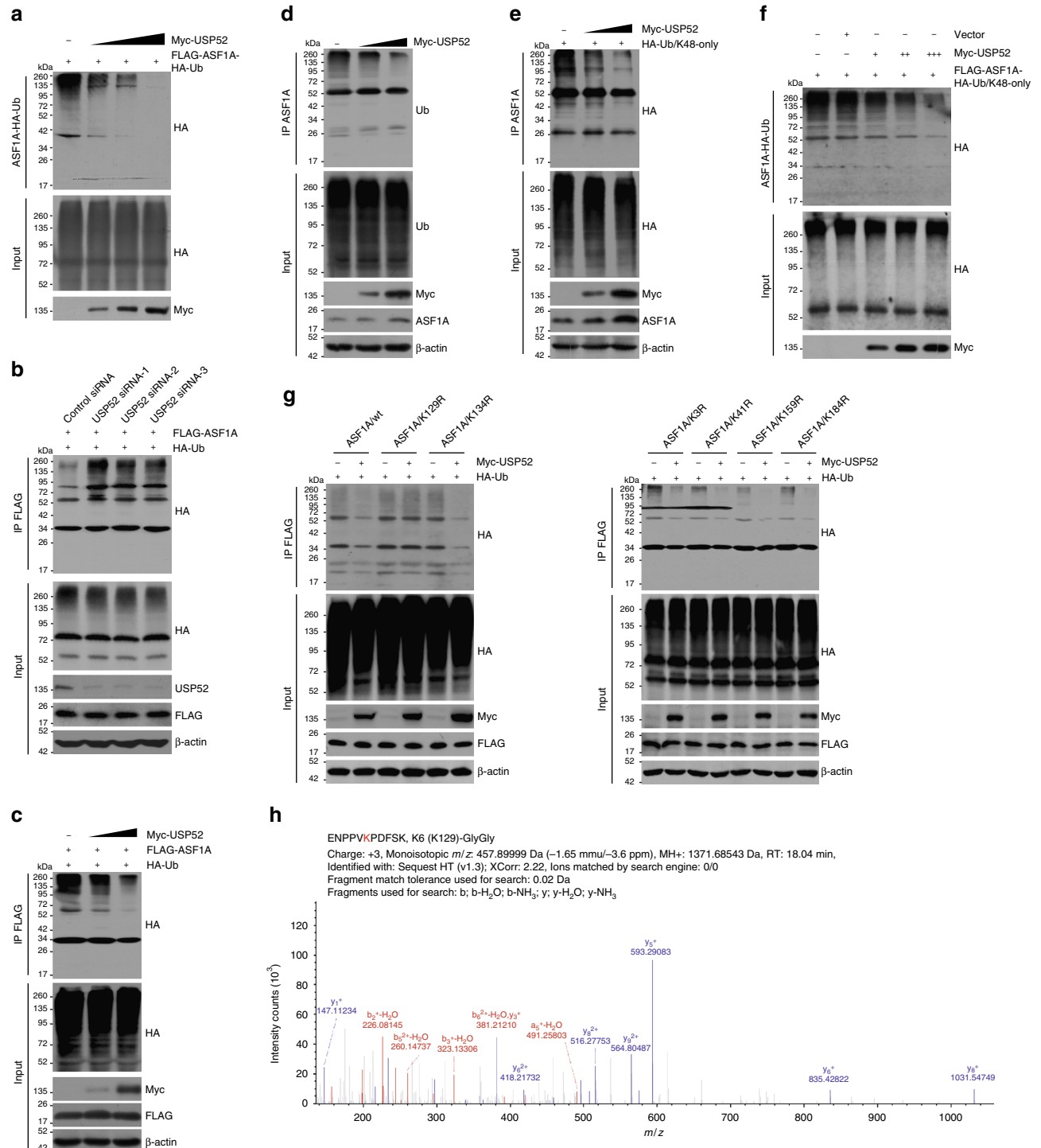

be associated with progression of multiple types of malignancies[18,49,59,60]. In light of our observation that USP52-mediated ASF1A stabilization regulates the level of H3K56Ac, it is reasonable to postulate that USP52/ASF1A signaling pathway plays a role in tumorigenesis. To test this hypothesis, we first analyzed the protein expression levels of USP52 and ASF1A with human tissue arrays including series of tumor samples with each type of cancer having three malignant samples paired with adjacent normal tissues. Immunohistochemical staining showed an upregulation of both USP52 and ASF1A in carcinomas derived from breast, kidney, and rectum within at least two of three paired samples (Supplementary Fig. 6).

Since H3K56Ac level has been reported to be elevated in breast cancer[18], we next analyzed, by immunohistochemical (IHC) staining, the expression profiles of USP52 and ASF1A with samples from different histologic types of breast carcinoma and histologically normal mammary tissues in tumor adjacent regions (Fig. 6a). Quantitation analysis of the stainings showed that the expression levels of USP52 and ASF1A were highly expressed in various types of breast carcinoma samples, and the levels of their expression strongly correlated with each other (Fig. 6b). Consistently, similar observation was obtained in breast cancer cell lines and normal human mammary epithelial cells (HMECs) (Fig. 4j). Collectively, these observations suggest that USP52-promoted ASF1A stabilization is potentially associated with breast carcinogenesis.

Supporting this deduction, colony formation assays showed that knockdown of USP52 severely impeded the colony formation of MCF-7 cells (Fig. 6c). Importantly, overexpression of ASF1A could, at least partially, offset the effect induced by USP52 knockdown (Fig. 6d). Moreover, USP52 overexpression-associated effects could be alleviated by ASF1A depletion (Fig. 6e). Similar results were also obtained when the growth of breast cancer cells was examined (Supplementary Fig. 7a). Consistently, the same is true when colony formation assays (Supplementary Fig. 7b, 7c and 7d) or growth examination experiments (Supplementary Fig. 7e) were performed with ZR-75-1 cells. These results are in favor of the role of USP52-promoted ASF1A stabilization in chromatin assembly thus cell cycle progression, and indicate that USP52/ASF1A axis is required for breast cancer cell proliferation.

To further establish the role of USP52/ASF1A in cancer cell proliferation and breast carcinogenesis, we transplanted two types of breast tumor, developed from MCF-7 cells infected with control lentivirus or lentivirus carrying USP52 shRNAs, onto the mammary fat pads of athymic mice. Tumor growth and mice weight were monitored over 6 weeks. Notably, in athymic mice

that received tumor transplant with USP52-depleted, the tumor growth was greatly suppressed (Fig. 6f). Western blotting and IHC analysis of the harvested tumors indicated that the expression level of ASF1A in USP52-depleted tumors markedly decreased (Fig. 6f). Collectively, these results point a role of the USP52-promoted ASF1A stabilization in promoting breast carcinogenesis.

**USP52/ASF1A signaling confers DNA damage tolerance.** Since ASF1A-facilitated H3K56Ac incorporation is important for proper restoration of chromatin structure at DNA damage sites thus genome stability[29] and ASF1A is required for cellular recovery from checkpoint arrest upon DNA damage[28], we thereby hypothesized that the elevated expression of USP52 and ASF1A in breast cancer potentially confers cellular resistance to genotoxic insults. To test this idea, MCF-7 cells with genetic features of ER positive, HER2 negative, and wild type for BRCA1/2 and TP53 genes were transfected with control siRNA, USP52 siRNA, or ASF1A siRNA followed by different doses of X-ray generated irradiation. Cell viability examination indicated cells with USP52 or ASF1A depletion were more sensitive to higher dose of IR treatment, albeit to variable extent (Fig. 7a).

To further consolidate the functional link between USP52 and ASF1A in this process, control MCF-7 cells or cells stably expressing ASF1A were transfected with control siRNA or USP52 siRNA, and cells were then exposed to IR. The results from cell viability assay demonstrated that forced expression of ASF1A was capable of rescuing the cellular sensitivity phenotype associated with USP52 knockdown (Fig. 7b). Meanwhile, western blotting analysis revealed that the upregulation of H3K56Ac trigged by IR was disrupted upon USP52 depletion, the effect of which could be restored by ASF1A gain of function (Fig. 7b). Similar observations were obtained in ZR-75-1 cells, whose genetic background is nearly identical to that of MCF-7 cells (Fig. 7c). Next, we demonstrated USP52 deficiency rendered breast cancer cells hypersensitive to camptothecin (CPT) (Fig. 7d), a type of DNA damaging agent impairing replication fork progression, while cells stably expressing ASF1A overcome the effect induced by USP52 depletion (Fig. 7e). Collectively, these results support the notion that USP52-promoted ASF1A stabilization confers cellular resistance of breast cancer cells to genotoxic insults.

**Discussion**

In this study, we demonstrated that the previously identified pseudo-deubiquitinase USP52 is a bona fide deubiquitinase and it

---

**Fig. 3** USP52 opposes ASF1A polyubiquitination. **a** In vitro deubiquitination assays with HeLa cells-purified Myc-USP52 (1, 2, and 3 μg for each lane) and HA-Ub-conjugated FLAG-ASF1A with high salt and detergent buffer. **b** HeLa cells stably expressing FLAG-ASF1A were co-transfected with control siRNA or USP52 siRNA together with HA-Ub as indicated. Cellular extracts were prepared for co-immunoprecipitation assays with anti-FLAG followed by IB with anti-HA. **c** HeLa cells stably expressing FLAG-ASF1A were co-transfected with HA-Ub and different amounts of Myc-USP52. Cellular extracts were prepared for co-immunoprecipitation assays with anti-FLAG followed by IB with anti-HA. **d** HeLa cells were transfected with different amounts of Myc-USP52 and cellular extracts were prepared for co-immunoprecipitation assays with anti-ASF1A followed by IB with antibody against ubiquitin. **e** Cellular extracts from HeLa cells expressing HA-Ub/K48-only and different amounts of Myc-USP52 were prepared for co-immunoprecipitation assays with anti-ASF1A followed by IB with anti-HA. **f** In vitro deubiquitination assays with HeLa cells-purified FLAG-tagged ASF1A-Ub/K48-only and different amounts of Myc-USP52 (1, 2, and 3 μg for each lane) with high salt and detergent buffer. **g** HeLa cells stably expressing wild-type ASF1A (ASF1A/wt) or different K to R mutants were co-transfected with HA-Ub and control vector or Myc-USP52 as indicated. Cellular extracts were prepared for co-immunoprecipitation assays with anti-FLAG followed by IB with anti-HA. **h** Mass spectrometry analysis of ASF1A ubiquitin conjugation sites. HeLa cells stably expressing FLAG-ASF1A were co-transfected with HA-Ub and USP52 siRNA. Cellular extracts were collected and sequentially purified with anti-FLAG affinity gel and HA affinity gel to enrich HA-Ub-conjugated ASF1A. After trypsinization, the retrieved peptides were subjected to mass spectrometry analysis. Fragmentation spectrums and parameters of the identified ASF1A peptides with di-Glycine remnant are shown. Detailed results from the mass spectrometric analysis are provided as Supplementary Data 2

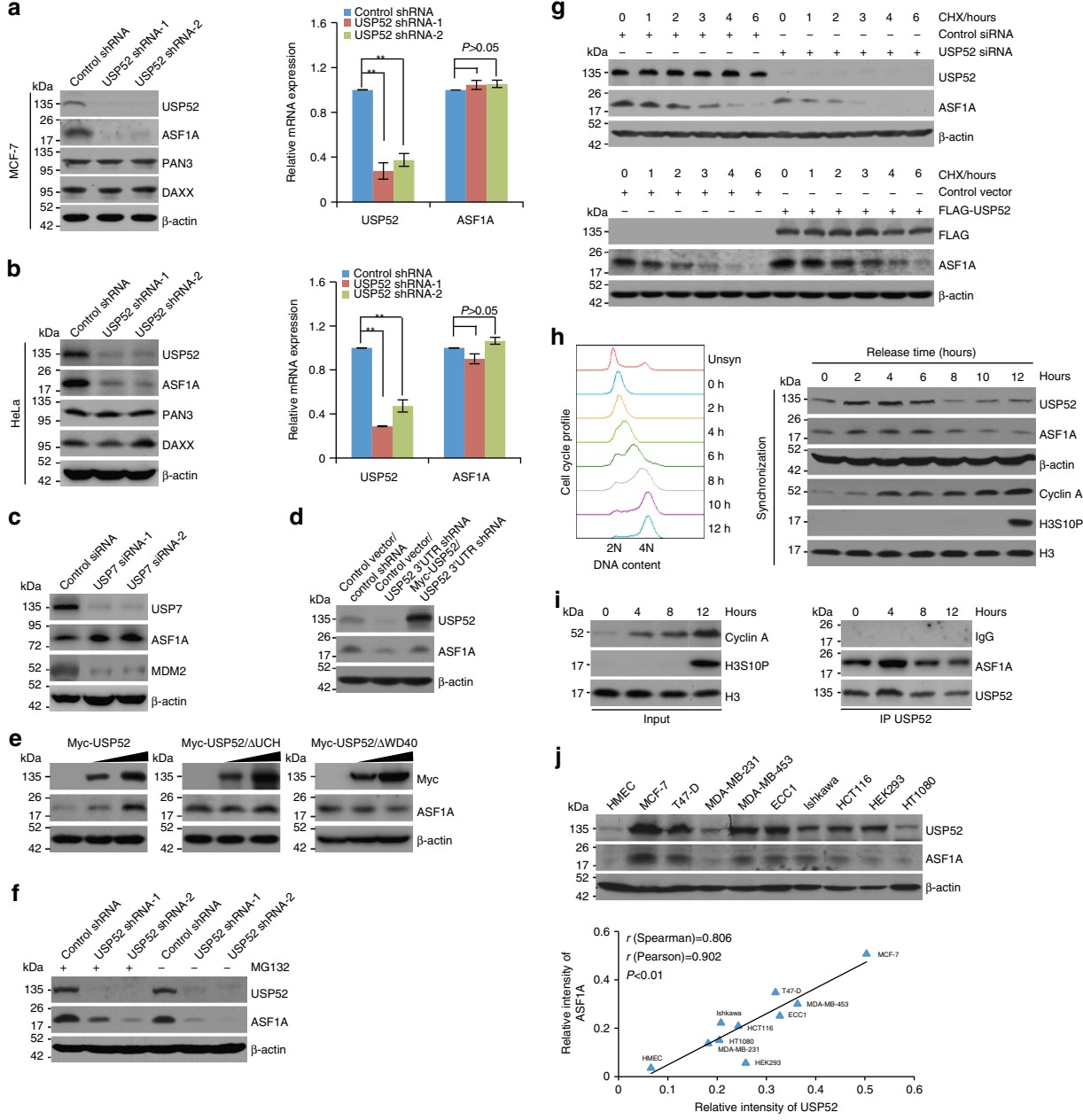

**Fig. 4** USP52 promotes ASF1A stabilization. **a** MCF-7 cells stably expressing different sets of USP52 shRNAs were collected for western blotting and qRT-PCR analysis. Each bar represents the mean ± S.D. for biological triplicate experiments. **P < 0.01, one-way analysis of variance (ANOVA). **b** Experiments analogous to **a** were performed in HeLa cells. Each bar represents the mean ± S.D. for biological triplicate experiments. **P < 0.01, one-way ANOVA. **c** MCF-7 cells were transfected with control siRNA or USP7 siRNA. Cellular extracts were prepared and analyzed by western blotting. **d** MCF-7 cells stably expressing shRNA targeting 3'UTR of USP52 were transfected with control vector or Myc-USP52 and cellular lysates were collected for western blotting analysis with antibodies against the indicated proteins. **e** MCF-7 cells were transfected with different amounts of wild-type USP52, USP52 lacking UCH domain (USP52/ΔUCH), or USP52 lacking WD40 repeat domain (USP52/ΔWD40), and cellular lysates were collected for western blotting analysis with antibodies against the indicated proteins. **f** MCF-7 cells stably expressing different sets of USP52 shRNAs were treated with proteasome inhibitor MG132 (10 μM) or DMSO. Cellular extracts were prepared and analyzed by western blotting. **g** MCF-7 cells transfected with control siRNA or USP52 siRNA were treated with cycloheximide (CHX) and harvested at the indicated time followed by western blotting analysis (upper panel). MCF-7 cells stably expressing control vector or FLAG-USP52 were treated with CHX and harvested at the indicated time followed by western blotting analysis (lower panel). **h** HeLa cells synchronized by double-thymidine block were released and cellular extracts were collected for western blotting analysis with antibodies against the indicated proteins. Representative cell cycle profiles are shown. **i** HeLa cells synchronized by double-thymidine block were released and cellular extracts were collected for co-immunoprecipitation analysis of the association of ASF1A with USP52. **j** Western blotting analysis of the expression of ASF1A and USP52 in multiple cell lines. The intensity of each band was quantified by densitometry with Image J software with β-actin as a normalizer. The correlation coefficient and P-values are shown

promotes ASF1A stabilization, at least, by removing K48-linked polyubiquitin chains. Immunoprecipitation analysis revealed that USP52 is preferentially associated with ASF1A in the S phase of the cell cycle, when ASF1A transfers H3.1–H4 dimer to replicating chromatin. We envisioned that USP52 interacts with and

stabilizes ASF1A at this time, thus supplies enough ASF1A to hand-off histones required for newly chromatin formation. Indeed, USP52 deficiency impaired chromatin assembly and DNA replication as well as S-phase progression, the effect of which could be largely restored by ASF1A gain of function.

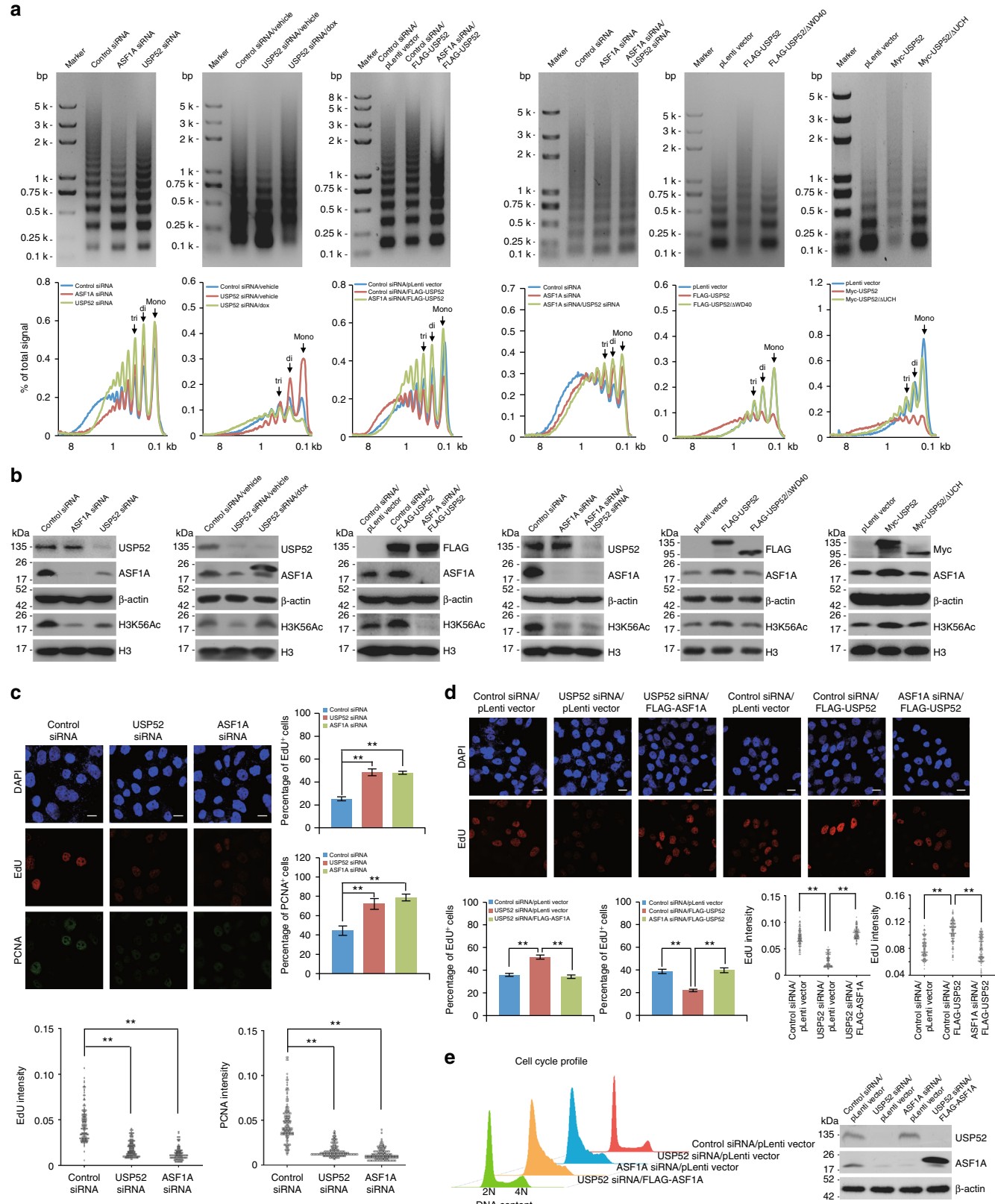

Remarkably, we also revealed that histone H3K56Ac levels were downregulated upon impairment of USP52-promoted ASF1A stabilization. Considering that ASF1A also plays an important role in replication-independent chromatin assembly and chromatin disassembly during transcriptional activation[2,16,54], it is interesting to investigate whether USP52-promoted ASF1A stabilization plays a role in these processes.

Unlike normal cells, cancer cells divide more often and therefore are more prone to undergo active DNA damage from replication fork stalling or collapse, the repair of which is essential for cancerous cells survival[61,62]. This feature makes DNA damage and response pathways as an ideal target for therapeutic intervention. Since ASF1A-regulated histone deposition is required for restoration of chromatin structures and recovery from replication stress as well as cell cycle arrest[28,29], upregulation of ASF1A together with H3K56Ac in multiple malignancies may endow cancerous cells with the capacity to efficiently counteract replication stress or exogenous DNA damage. In this study, we revealed that the expression level of USP52 and ASF1A is upregulated in breast cancer and positively correlates with each other, and, indeed, USP52-promoted ASF1A stabilization is required for breast cancer cells to combat with genotoxic insults. Our findings suggest that inhibition of the enzymatic activity of USP52 or disruption of the association of USP52 with ASF1A, combined with chemo- or radio-therapy, could potentially improve the treatment of breast cancer.

IHC analysis of multiple tumor and adjacent tissues revealed that the expression levels of USP52 and ASF1A are both upregulated in breast and rectum tumors. This observation agrees well with the previous finding that ASF1A-deposited histone mark H3K56Ac is elevated in these tumors[18]. Although the expression level of ASF1A is upregulated in lung cancer as reported[49], the protein abundance of USP52 appears to be downregulated. A recent study showed that E2 ubiquitin-conjugating enzyme RAD6 and E3 ligase MDM2 cooperatively promote ASF1A degradation[49], thus we envision that downregulation of RAD6 or/and MDM2 could possibly lead to ASF1A upregulation in lung cancer even with less abundance of USP52. Similarly, this deduction could also be applied to interpret the non-correlated expression profiles of USP52 and ASF1A in other types of tumors. Although transcriptome analysis indicated that the mRNA expression level of ASF1A in breast tumor is similar to that in normal breast tissues[27], our observations showed that the protein abundance of ASF1A is upregulated due to elevated expression of its protein deubiquitinase USP52 in breast cancer. Since complicated heterogeneity observed in breast cancer, it will, however, be important to determine to what extent USP52 and/or ASF1A affect tumor survival across genetically distinct subgroups of breast cancer.

In previous studies, USP52 has been considered as a pseudo-deubiquitinase[41,42]. Surprisingly, we demonstrate that Sf9 cells-purified UCH domain of USP52 (USP52/UCH), but not that from bacteria, is capable of trimming K6-, K11-, K48-, K63-, and M1-linked ubiquitin chains with a preference for the K63-linked ubiquitin moieties (Figs. 2c, e, f). We also noted that USP52/UCH prefers to cleave shorter ubiquitin linkages. In particular, K48-linked di-ubiquitins could be more efficiently hydrolyzed than K48-linked tetra-ubiquitins (Fig. 2c). However, it appears that full-length USP52 prefers longer ubiquitin chains (K48-linked) conjugated on ASF1A and displays higher catalytic activity than USP52/UCH, while K63-linked ubiquitin species conjugated on ASF1A is resistant to cleavage by USP52. The discrepancies observed above could likely be attributed to factors such as intra-protein collaborations, protein–protein interactions, post-translational modifications, or a combination of these, which could potentially act to enhance USP52 enzymatic activity or impact on its topology preference of polyubiquitin chains. Similar to our observations, recent studies reported that the deubiquitinase MINDY-1 catalytic domain is less efficient in cleaving long K48-polyubiquitin chains compared to the full-length deubiquitinase[63]. Another pertinent example supporting our deduction comes from the understanding of the catalytic activity modulation of deubiquitinase USP7, whose C-terminal ubiquitin-like (UBL) domain markedly promotes its deubiquitinating activity and this effect could be further allosterically activated by the metabolic enzyme GMP-synthetase (GMPS)[64]. Since differently linked ubiquitin polymers have distinct cellular functions[45,50], it will be interesting to investigate whether USP52 is capable of cleaving K6-, K11-, or M1-linked ubiquitin chains conjugated onto ASF1A, the elucidation of which together with the correspondingly biological effects will provide more useful information on understanding of the functional link between USP52 and ASF1A.

Interestingly, the conserved cysteine residue and histidine residue in the triad of classical DUBs are replaced by alanine (A526) and serine (S867), respectively, in the UCH domain of USP52 (Supplementary Fig. 2a). We hypothesized that particular post-translational modifications or protein folding could rearrange C528 or C530 and histidine adjacent to S867, like H871, to form newly active center thus enable the UCH domain to hydrolyze ubiquitin chains. Although NEM treatment almost completely abolished the catalytic activity of USP52 against tetra-ubiquitins or polyubiquitinated ASF1A, mutational analysis showed that C528 and C530 are not critical for the catalytic

**Fig. 5** USP52 promotes chromatin assembly through stabilizing ASF1A. **a** MNase digestion assay. Control HeLa cells or HeLa cells stably expressing the indicated genes were transfected with the indicated siRNAs and synchronized by nocodazole followed by drug removal and FACS analysis at 2 h interval. Cell cycle profiles after synchronization and release into specific cell cycle stages are shown in Supplementary Fig. 5a. The same amounts of isolated nuclei from S-phase cells were treated with MNase (0.4 gel Unit/μl) and purified DNAs were resolved in 1.5% agarose gels followed by EtBr staining. The MNase digestion patterns were quantified by densitometry and the positions of mono-, di-, and tri-nucleosomal DNA fragment are indicated. In panel 2, cells with Dox-inducible expression of ASF1A were transfected with control siRNA or USP52 siRNA in the presence or absence of doxycycline. **b** Cellular lysates from cells in **a** were analyzed by western blotting with antibodies against the indicated proteins. **c** Immunofluorescence analysis of HeLa cells transfected with control siRNA, USP52 siRNA, or ASF1A siRNA followed by EdU pulse labeling. PCNA staining served as a marker for S-phase cells. Scale bar, 10 μm. Percentage of EdU or PCNA-positive cells with both strong and weak stainings was counted. Each bar represents the mean ± S.D. for biological triplicate experiments. The distribution of EdU or PCNA intensities from more than 200 cells is presented as dot plotting. ***P* < 0.01, one-way ANOVA. **d** HeLa cells stably expressing FLAG-ASF1A or FLAG-USP52 were transfected with control siRNA, USP52 siRNA, or ASF1A siRNA as indicated and analyzed by immunofluorescence after EdU pulse labeling. Scale bar, 10 μm. Percentage of EdU-positive cells with both strong and weak stainings was counted. Each bar represents the mean ± S.D. for biological triplicate experiments. The distribution of EdU intensities from almost 200 cells is presented as dot plotting. ***P* < 0.01, one-way ANOVA. **e** Control cells or HeLa cells stably expressing FLAG-ASF1A were transfected with indicated siRNAs and cell cycle profiles were analyzed by FACS. Cellular lysates from these cells were analyzed by western blotting with antibodies against the indicated proteins

activity of USP52. In the future, functional analysis of distal or other proximal cysteine/histidine residue mutants with deubiquitination assays, and crystal structure analysis of the UCH domain of USP52 purified from Sf9 cells together with different

types of ubiquitin linkages, will be helpful in characterizing the essential residues required for catalytic activity and understanding the molecular mechanism of USP52 hydrolyzing the isopeptide bond of distinct ubiquitin linkages. On the other side, cysteine

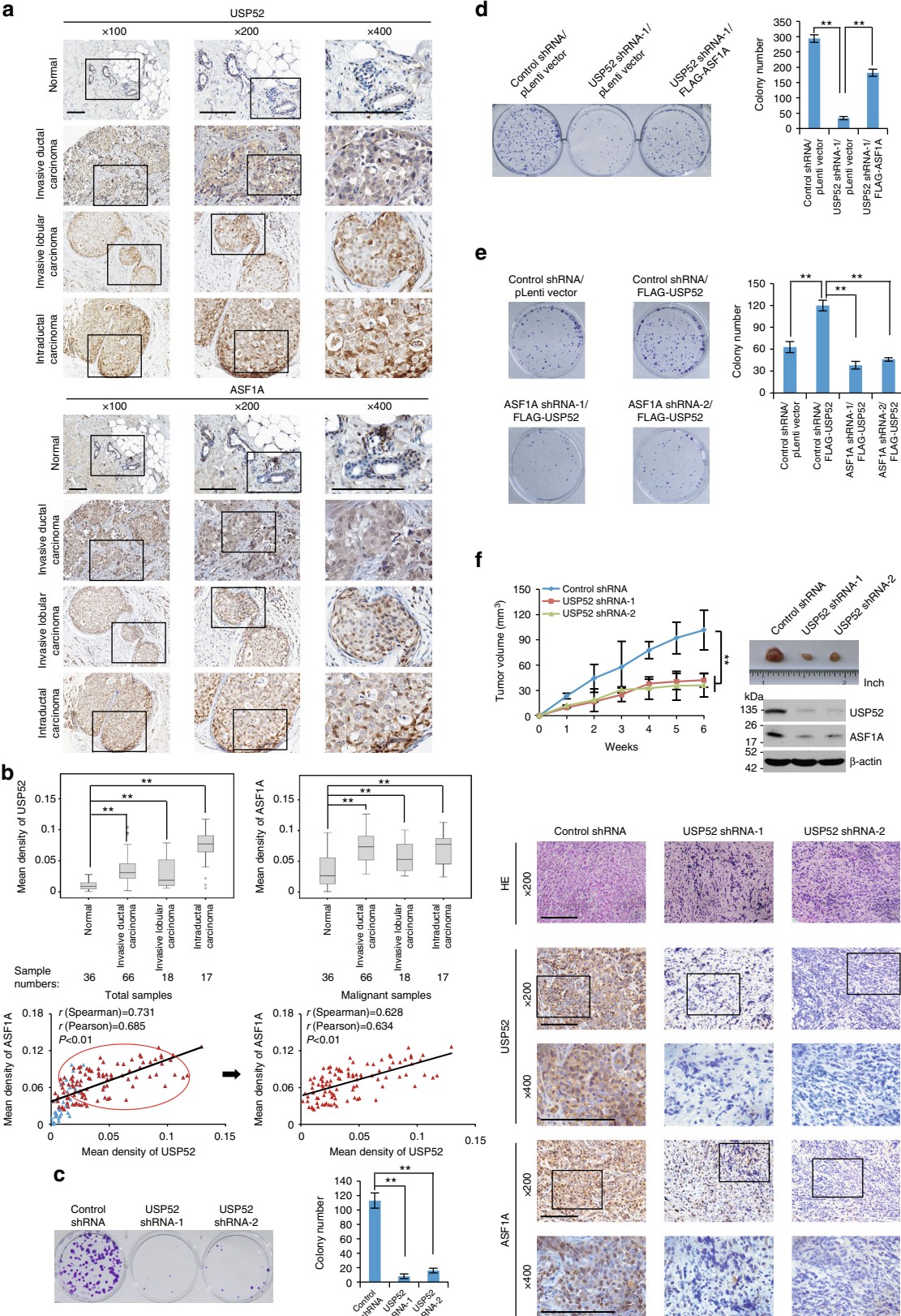

residues might contribute to, but are not essential for the catalytic activity of certain DUBs. A recent study reported that SidJ, an effector protein from the bacterial pathogen *Legionella pneumophila*, is able to hydrolyze K11-, K33-, K48-, and K63-linked diubiquitins without requirement of catalytic cysteine residues, although the enzymatic activity of SidJ is sensitive to NEM treatment[48]. Likewise, USP52-catalyzed deubiquitination could be achieved by a mechanism different from those used by classical DUBs. Nevertheless, our study identified that USP52 is a bona fide USP, and revealed that USP52 deubiquitinates and stabilizes histone chaperone ASF1A.

## Methods

**Antibodies and reagents**. The sources of antibodies against the following proteins were: HA (sc-805, 1:500 for WB) from Santa Cruz Biotechnology; β-actin (A1978, 1:10,000 for WB) and FLAG (F3165, 1:10,000 for WB) from Sigma; ASF1A (ABE149, IP, 1:1000 for WB and 1:100 for IHC), ASF1A (MABE90, 1:200 for IF) and Histone H3S10P (05-1336, 1:1000 for WB) from Millipore; Histone H3K56Ac (39281, 1:1,000 for WB) from Active Motif; USP7 (A300-033A, 1:1000 for WB) from Bethyl; DAXX (ab105173, 1:1000 for WB), PAN3 (ab88642, 1:1000 for WB) and Histone H3 (ab1791, 1:10,000 for WB) from Abcam; USP52 (16427-1-AP, IP, 1:1000 for WB, 1:100 for IHC and 1:200 for IF), Cyclin A (18202-1-AP, 1:1000 for WB), His (10001-0-AP, 1:2000 for WB), MDM2 (19058-1-AP, 1:1000 for WB) and PCNA (60097-1-Ig, 1:100 for IF) from Proteintech; Myc (M047-3, 1:10,000 for WB) from MBL; ubiquitin (OM294553, 1:500 for WB) from OmnimAbs; and ubiquitin (RLT4793, 1:500 for WB) from Ruiying Biological. Anti-HA affinity gel (E6779), Anti-FLAG M2 affinity gel (A2220), 3× FLAG peptide (F4799), Anti-c-Myc agarose affinity gel (A7470), Myc peptide (M2435), thymidine (T1895), NEM (E3876), MG132 (SML1135), and doxycycline (D9891) were purchased from Sigma. K6-linked tetra-ubiquitin chains (UC-15), K11-linked tetra-ubiquitin chains (UC-45), K29-linked tetra-ubiquitin chains (UC-83), K33-linked tetra-ubiquitin chains (UC-103), K48-linked tetra-ubiquitin chains (UC-210B), K63-linked tetra-ubiquitin chains (UC-310B), M1-linked tetra-ubiquitin chains (UC-710B), K2-linked di-ubiquitin chains (UC-61B) and K48-linked di-ubiquitin chains (UC-200B) were purchased from Boston Biochem. CHX (0970, working concentration 50 μg/ml) was purchased from TOCRIS. MNase (M0247S) was purchased from New England Biolabs.

**Plasmids**. The FLAG-tagged ASF1A were carried by pLenti-Tight-Puro vector or pLenti-Hygro vector, while the FLAG-tagged ASF1B were carried by pLenti-Hygro vector. The HA-FLAG-tagged H3.1 and H3.3 were carried by pcDNA3.1 vector. The FLAG-tagged full length and truncation mutants of ASF1A were carried by pcDNA3.1 vector (for in vitro transcription/translation). The K3R, K41R, K129R, K134R, K159R and K184R ASF1A mutants carried by pLenti-Hygro were constructed by quick change strategy using point mutation kit from Stratagene. The FLAG-GFP-tagged full length and truncation mutants of USP52 were carried by pcDNA3.1 vector (for transient transfection or in vitro transcription/translation). The FLAG-tagged full length and truncation mutants of USP52 were carried by pLenti-Hygro vector (for stably expressing in cells). Myc-tagged truncation mutants and full length of USP52 were carried by pLenti-Hygro vector. Full length or truncations of GST fusion ASF1A, USP52/WD40, and USP52/UCH were amplified and integrated into pGEX-4T-3 vector. USP52/UCH and C528A/C530A mutant expressed in insect cells were carried by pFastBac-HTA vector. HA-tagged ubiquitin K48-only (Plasmid #17605, Addgene) and K63-only (Plasmid #17606, Addgene) were gifts from Dr. Ted Dawson (Johns Hopkins University School of Medicine, Baltimore).

**Cell culture**. MCF-7, ZR-75-1, U2OS, HeLa, HEK293, HEK293T, MDA-MB-231, MDA-MB-453, HCT116, T47-D, HT1080, HMEC and Sf9 cells were got from the American Type Culture Collection (Manassas, VA) and cultured under the manufacturer's instructions. Ishkawa and ECC1 cells were kindly provided by Dr. Myles Brown (Dana-Farber Cancer Institute, Boston) and cultured in RPMI 1640 medium with 10% of fetal bovine serum (FBS, Biological Industries). Cells that allow protein expression under doxycycline treatment were created in two steps. First, cells were infected with lentivirus carrying rtTA and subjected to Neomycin selection. Subsequently, the established rtTA cells were infected with virus carrying pLenti-Tight-Puro vector that encodes ASF1A, followed by puromycin selection. All of the cells integrated with rtTA were cultured in Tet Approved FBS and medium from Clontech. All of the cells were authenticated by examination of morphology and growth characteristics, and were confirmed to be mycoplasma-free.

**Western blotting**. Whole-cell lysates were harvested from treated cells followed by re-suspending in 5× SDS-PAGE loading buffer. The boiled protein samples were then subjected to SDS-PAGE followed by immunoblotting with appropriately primary antibodies and secondary antibodies. Uncropped scans for main figure blots with associated markers and molecular weights are provided in the Supplementary Information (Supplementary Fig. 8).

**Immunopurification and silver staining**. Lysates from MCF-7 cells stably expressing FLAG-ASF1A were prepared by incubating the cells in lysis buffer containing protease inhibitor Cocktail (Roche). Anti-FLAG immunoaffinity columns were prepared using anti-FLAG M2 affinity gel (Sigma) following the manufacturer's suggestions. Cell lysates were obtained from about $5 \times 10^8$ cells and applied to an equilibrated FLAG column of 1 ml bed volume to allow for adsorption of the protein complex to the column resin. After binding, the column was washed with cold phosphate-buffered saline (PBS) plus 0.2% Nonidet P-40. FLAG peptide (Sigma) was applied to the column to elute the FLAG protein complex as described by the vendor. The eluents were collected and visualized on NuPAGE 4–12% Bis-Tris gel (Invitrogen) followed by silver staining with silver staining kit (Pierce). The distinct protein bands were retrieved and analyzed by LC-MS/MS.

**Nano-HPLC-MS/MS analysis of ASF1A-containing protein complex**. To identify proteins associated with FLAG-ASF1A, LC-MS/MS analysis was performed using a Thermo Finnigan LTQ linear ion trap mass spectrometer in line with a Thermo Finnigan Surveyor MS Pump Plus HPLC system. Tryptic peptides generated were loaded onto a trap column (300SB-C18, $5 \times 0.3$ mm, 5 μm particle; Agilent Technologies, Santa Clara, CA) which was connected through a zero dead volume union to the self-packed analytical column (C18, 100 μm i.d. × 100 mm, 3 μm particle; SunChrom, Germany). The peptides were then eluted over a gradient (0–45% B in 55 min, 45–100% B in 10 min, where B = 80% Acetonitrile, 0.1% formic acid) at a flow rate of 500 nL/min and introduced online into the linear ion trap mass spectrometer (Thermo Fisher Corporation, San Jose, CA) using nano electrospray ionization. Data-dependent scanning was incorporated to select the five most abundant ions (one microscan per spectra; precursor isolation width 1.0 $m/z$, 35% collision energy, 30 ms ion activation, exclusion duration: 90 s; repeat count: 1) from a full-scan mass spectrum for fragmentation by collision-induced dissociation. MS data were analyzed using SEQUEST (v. 28) against NCBI human protein database (14 December 2011 downloaded, 33,256 entries), and results were filtered, sorted, and displayed using the Bioworks 3.2. Peptides (individual spectra) with preliminary score (Sp) ≧500; rank of Sp (RSp) ≦5; and peptides with +1, +2, or +3 charge states were accepted if they were fully enzymatic and had a cross-correlation (Xcorr) of 1.90, >2.75, and >3.50, respectively. The following residue modifications were allowed in the search: carbamidomethylation on cysteine as fix modification and oxidation on methionine as variable modification. Peptide sequences were searched using trypsin specificity and allowing a maximum of two missed cleavages. Sequest was searched with a peptide tolerance of 3.0 Da and a fragment ion tolerance of 1.0 Da.

**Fig. 6** USP52/ASF1A signaling axis is implicated in breast carcinogenesis. **a** Immunohistochemistry analysis of the expression levels of USP52 and ASF1A in different histologic types of breast tumors and adjacent normal mammary tissues. Representative images (×100, ×200, and ×400 magnification) from these samples are shown. Scale bar, 50 μm. **b** Scores of the stained sections from **a** were determined by evaluating the intensity of immunopositivity by Image-pro Plus software and are presented with box plots. **P < 0.01, one-way ANOVA. The correlation coefficient and *P*-values were analyzed as indicated. **c** Colony formation assays of MCF-7 cells stably expressing USP52 shRNAs. Representative images from biological triplicate experiments are shown. **P < 0.01, one-way ANOVA. **d** Colony formation assays of MCF-7 cells stably expressing the indicated genes and shRNAs. Representative images from biological triplicate experiments are shown. **P < 0.01, one-way ANOVA. **e** Colony formation assays of MCF-7 cells stably expressing the indicated genes and shRNAs. Representative images from biological triplicate experiments are shown. **P < 0.01, one-way ANOVA. **f** MCF-7 tumors with the indicated treatments were transplanted onto athymic mice and tumor volumes were measured weekly. Each bar represents the mean ± S.D. for different animal measurements (n = 6). **P < 0.01, two-way ANOVA. The levels of USP52 and ASF1A proteins in these tumors were examined by western blotting and immunohistochemistry analysis. Representative images (×200 and ×400 magnification) are shown. Scale bar, 50 μm

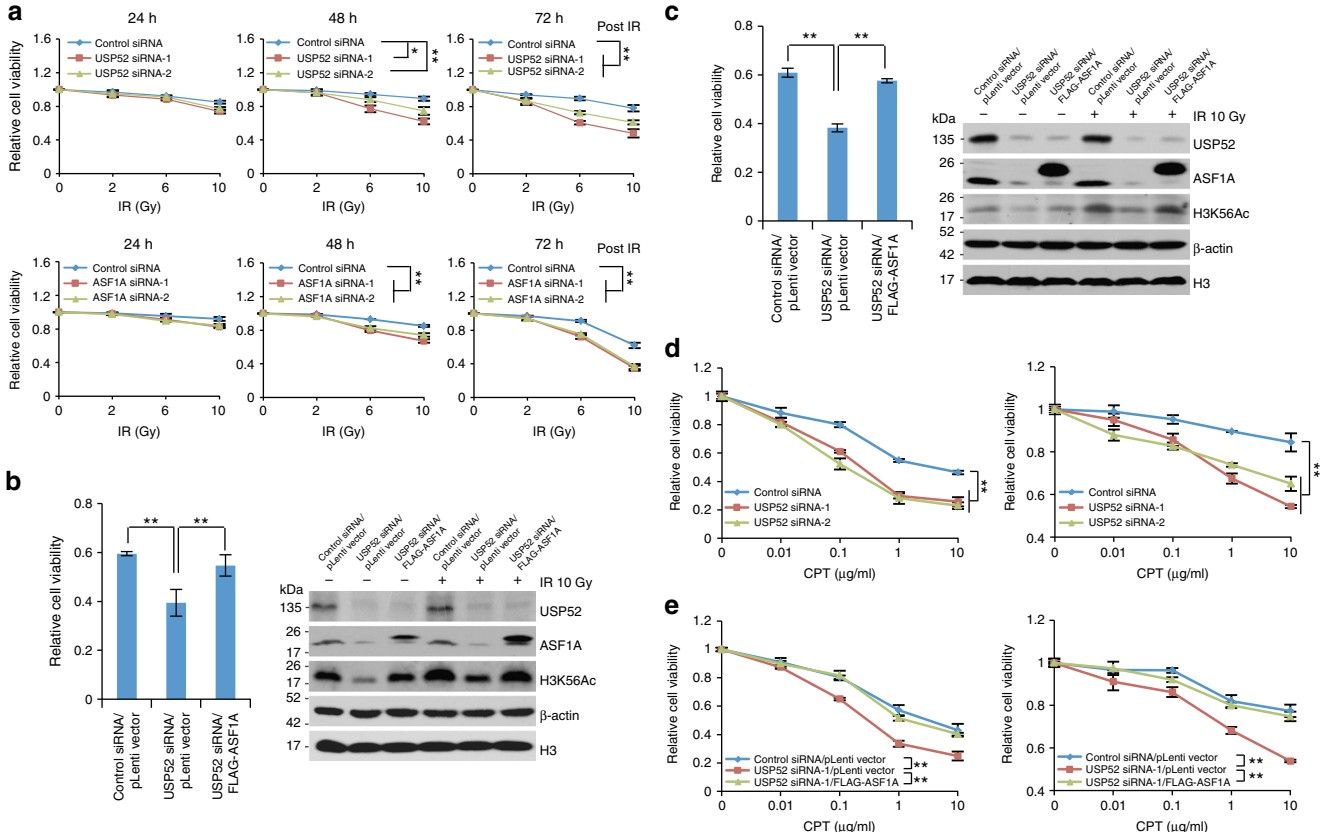

**Fig. 7** USP52/ASF1A signaling axis confers DNA damage tolerance. **a** MCF-7 cells transfected with control siRNA, USP52 siRNA, or ASF1A siRNA were exposed to increasing dosage of IR and cell viability was examined at different time points. Each bar represents the mean ± S.D. for biological triplicate experiments. $^*P < 0.05$, $^{**}P < 0.01$, two-way ANOVA. **b** Control MCF-7 cells or cells stably expressing ASF1A were transfected with control siRNA, or USP52 siRNA. Then cells were exposed to 10 Gy of IR and cell viability was examined. Relative cell viability was calculated with control treatment as a normalizer and each bar represents the mean ± S.D. for biological triplicate experiments. $^{**}P < 0.01$, one-way ANOVA. Cellular extracts were collected for western blotting analysis. **c** Experiments analogous to **b** were performed with ZR-75-1 cells. $^{**}P < 0.01$, one-way ANOVA. **d** MCF-7 cells (left panel) or ZR-75-1 cells (right panel) transfected with control siRNA or USP52 siRNA were treated with CPT at indicated dosage and cell viability was examined. Each bar represents the mean ± S.D. for biological triplicate experiments. $^{**}P < 0.01$, two-way ANOVA. **e** MCF-7 cells (left panel) or ZR-75-1 cells (right panel) stably expressing ASF1A were transfected with control siRNA or USP52 siRNA. Then cells were treated with CPT at indicated dosage and cell viability was examined. Each bar represents the mean ± S.D. for biological triplicate experiments. $^{**}P < 0.01$, two-way ANOVA

**LC-MS/MS analysis of ASF1A ubiquitination sites**. FLAG-ASF1A-conjugated ubiquitin bands were excised from the gel and subjected to in-gel tryptic digestion. Resulting peptides were separated by reverse-phase liquid chromatography on an easy-nLC 1000 system (Thermo Fisher Scientific) and directly sprayed into a Q-Exactive Plus mass spectrometer (Thermo Fisher Scientific). The mass spectrometry analysis was carried out in a data-dependent mode with an automatic switch between a full MS and an MS/MS scan in the obitrap. For full MS survey scan, automatic gain control target was 3e-6, and scan range was from 350 to 1800 with a resolution of 70,000. The 10 most intense peaks with charge state ≥2 were selected for fragmentation by higher-energy collision dissociation with normalized collision energy of 27%. The MS2 spectra were acquired with 17,500 resolution. The exclusion window was set at ±1.6 Da. All MS/MS spectra were searched against the Uniprot-Human protein sequence database by using Proteome Discoverer software (v1.4) with an overall false discovery rate for peptides of less than 1%. Peptide sequences were searched using trypsin specificity and allowing a maximum of two missed cleavages. Carbamidomethylation on Cys was specified as fixed modification. Gly–Gly on lysine, oxidation of methionine, and acetylation on peptide N-terminal were fixed as variable modifications. Mass tolerances for precursor ions were set at ±10 ppm for precursor ions and ±0.02 Da for MS/MS. The interesting MS/MS spectra were manually verified.

**Fast protein liquid chromatography chromatography**. FLAG-ASF1A-containing protein complexes were applied to a Superose 6 size exclusion column (GE Healthcare) that had been equilibrated with dithiothreitol-containing buffer and

calibrated with protein standards (Amersham Biosciences). The column was eluted at a flow rate of 0.5 ml/min and fractions were collected every 2 min.

**Immunoprecipitation**. Cellular lysates were prepared by incubating the cells in NETN buffer (50 mM Tris-HCl, pH 8.0, 150 mM NaCl, 0.2% Nonidet P-40, 2 mM EDTA) in the presence of protease inhibitor Cocktails (Roche) for 20 min at 4 °C followed by centrifugation at 14,000 g for 15 min at 4 °C. For immunoprecipitation, about 500 µg of protein was incubated with control or specific antibodies (1–2 µg) for 12 h at 4 °C with constant rotation; 50 µl of 50% protein G magnetic beads (Invitrogen) was then added and the incubation was continued for an additional 2 h. Beads were then washed five times using the lysis buffer. Between washes, the beads were collected by magnetic stand (Invitrogen) at 4 °C. The precipitated proteins were eluted from the beads by re-suspending the beads in 2× SDS-PAGE loading buffer and boiling for 5 min. The boiled immune complexes were subjected to SDS-PAGE followed by immunoblotting with appropriate antibodies.

**Recombinant protein purification**. Recombinant baculovirus carrying mutants of USP52 was generated with the Bac-to-Bac System (Invitrogen). Infected Sf9 cells were grown in spinner culture for 48–96 h at 27 °C and lysed by ultrasonicator in Equilibration buffer (50 mM sodium phosphate, 0.3 M sodium chloride, 10 mM imidazole, and 10 mM Tris-HCl, pH 8.0). His-tagged proteins were purified using $Ni^{2+}$-NTA agarose (Invitrogen) according to the standard procedures. GST tagged full length or deletion mutants of ASF1A and USP52 were purified from bacteria BL21 cells with Glutathione-agarose.

**In vitro deubiquitination assay**. HeLa cells expressing full-length ASF1A and HA-ubiquitin were collected and then lysed in RIPA Buffer (300 mM NaCl, 0.5% sodium deoxycholate, 0.1% SDS, 1% Nonidet P-40, and 50 mM Tris-HCl, pH 8.0). The resulting lysate was incubated with anti-FLAG affinity gel for 2 h and the beads were then washed five times with RIPA Buffer, eluted with $3 \times$ FLAG peptide and then subjected to HA affinity gel to enrich HA-Ub-conjugated ASF1A (ASF1A-Ub). HeLa cells expressing full-length USP52 were collected and then lysed in RIPA buffer. The resulting lysate was incubated with anti-Myc or anti-FLAG affinity gel for 2 h and the beads were then washed five times with RIPA buffer followed by excess peptides elution. Recombinant USP52 and ASF1A-Ub were incubated in DUB buffer (50 mM HEPES, pH 7.5, 10 mM 2-mercaptoethanol and 0.5 mM EDTA) at 37 °C for 4 h. The reactions were stopped by boiling for 5 min in 5× SDS-PAGE loading buffer followed by western blotting with appropriate antibodies. Analogously, recombinant USP52 mutants were incubated with different types of homogeneous ubiquitin linkages in DUB buffer followed by western blotting analysis.

**In vivo deubiquitination assay**. Cells with different treatments were lysed in RIPA buffer in the presence of protease inhibitors at 4 °C for 30 min with rotation, and centrifuged at 20,000 g for 15 min. About 0.5–1.5 mg of cellular extracts were immunoprecipitated with anti-FLAG affinity gel for 2 h. The beads were then washed five times with RIPA buffer, boiled in SDS loading buffer, and subjected to SDS-PAGE followed by immunoblotting.

**RNA interference**. All siRNA transfections were performed using Lipofectamine RNAiMAX (Invitrogen) following the manufacturer's recommendations. The final concentration of the siRNA molecules is 10 nM and cells were harvested 72 or 96 h later according to the purposes of the experiments. Control siRNA (ON-TAR-GETplus Non-Targeting Pool, D-001810-10), ASF1A siRNA (ON-TARGETplus, M-020222-01-0005) and USP52 siRNA (ON-TARGETplus, L-021192-00-0005) were got from Dharmacon in a smart pool manner, while the individual siRNAs against USP52, ASF1A, and USP7 were chemically synthesized by Sigma (Shanghai, China). The shRNAs against USP52 were purchased from Sigma and the DNA oligos targeting Luciferase or ASF1A were carried by pLL3.7 lentiviral vector. The sequences of siRNAs and shRNAs are provided in Supplementary Table 1 and Supplementary Table 2, respectively.

**qRT-PCR**. Total cellular RNAs were isolated with TRIzol reagent (Invitrogen) and used for first-strand cDNA synthesis with the Reverse Transcription System (Roche). Quantitation of all gene transcripts was done by qPCR using a Power SYBR Green PCR Master Mix (Roche) and an ABI PRISM 7500 sequence detection system (Applied Biosystems) with the expression of *ACTB* as the internal control. The primers used are listed in Supplementary Table 3.

**Lentiviral production**. The shRNAs targeting USP52 or ASF1A or vectors encoding rtTA, USP52, and ASF1A, as well as three assistant vectors: pMDLg/pRRE, pRSV-REV, and pVSVG, were transiently transfected into HEK293T cells. Viral supernatants were collected 48 h later, clarified by filtration, and concentrated by ultracentrifugation.

**Immunofluorescence**. MCF-7 Cells on glass coverslips (BD Biosciences) were fixed with 2% paraformaldehyde and permeabilized with 0.2% Triton-X-100 in PBS. Samples were then blocked in 5% donkey serum in the presence of 0.1% Triton-X-100 and stained with the appropriate primary and secondary antibodies coupled to AlexaFluor 488 or 594 (Invitrogen). Confocal images were captured on FluoView1000 Olympus using a ×60 oil objective. To avoid bleed-through effects in double-staining experiments, each dye was scanned independently in a multi-tracking mode.

**Cell flow cytometry**. Cells with different treatments were trypsinized, washed with PBS, and fixed in ice-cold 70% ethanol at 4 °C overnight. After being washed with PBS, cells were incubated with RNAase A (Sigma) in PBS for 30 min at 37 °C and then stained with 50 mg/ml propidium iodide. Cell cycle data were collected with FACS Calibur (Becton Dickinson) and analyzed with FlowJo software. Apoptosis of cells was analyzed with FACS using Cells Annexin V Apoptosis Detection Kits as per the manufacture's standard procedures (Affymetrix eBioscience).

**MNase digestion assay**. About one million cells with indicated treatment were collected by ice-cold PBS, and the cell pellets were suspended in ice-cold lysis buffer (10 mM Tris-HCl, pH 7.5, 10 mM NaCl, 3 mM $MgCl_2$ and 0.4% Nonidet P-40) in the presence of protease inhibitors and incubated on ice for 5 min. The lysate was cleared with centrifugation at 2000 × g for 5 min at 4 °C, and then collected and washed with lysis buffer twice. The nuclear pellet was suspended with 50 μl glycerol buffer (10 mM Tris-HCl, pH 7.5, 0.1 mM EDTA, 5 mM MgAc₂ and 25% (vol/vol) glycerol), mixed with equal volume of 2 × MNase buffer (50 mM KCl, 8 mM $MgCl_2$, 2 mM $CaCl_2$, and 100 mM Tris-HCl, pH 7.5) and incubated at 37 °C for 5

min with 40 gel Unit of MNase (NEB) per 100 μl of total reaction volume. The reaction was stopped by adding EDTA to a final concentration of 10 mM. Genomic DNA was purified and separated by electrophoresis in 1.5% agarose gel.

**EdU incorporation assay**. HeLa cells were incubated with EdU in medium (50 μM) for 2 h. Then, the cells were fixed in 4% paraformaldehyde for 10 min, permeabilized with 0.5% Triton-X-100, and stained with Apollo® fluorescent dye, according to the manufacturer's instructions of the Cell-Light EdU DNA cell proliferation kit from RiboBio. Photographs of the cells were scanned independently in a multi-tracking mode with an OLYMPUS confocal microscopy.

**Colony formation assay**. MCF-7 or ZR-75-1 cells stably expressing indicated genes or/and shRNAs were maintained in culture media for 14 days. After 14 days, the cells were washed with PBS, fixed with methyl alcohol for 10 min, and stained with crystal violet (0.5% wt/vol) for 20 min. The number of colonies per well was counted.

**Tissue specimens**. The samples of carcinomas and the adjacent normal tissues were obtained from surgical specimens from patients with breast cancer or others. Samples were frozen in liquid nitrogen immediately after surgical removal and maintained at −80 °C. Prepared tissues were incubated with antibodies against USP52 or ASF1A and processed for immunohistochemistry with standard DAB staining protocols. Representative images for normal and malignant breast tumor samples are collected in different magnification fields. Images for tumor adjacent normal tissue (36), invasive ductal breast carcinoma (66), invasive lobular breast carcinoma (18), and intraductal breast carcinoma (17) samples were collected under microscopy with different magnifications. The image quality was evaluated and the background with uneven illumination was corrected with Image-Pro plus software. Then, the mammary ductal or lobular cells or carcinoma cells were selected as region of interest according to morphology features of the tissue or cells. The scores of the stained sections were determined by evaluating the extent and intensity of immunopositivity by Image-pro Plus software. Values that are less than or equal to the first quartile minus 1.5 times the interquartile range, or are greater than or equal to the third quartile plus 1.5 times the interquartile range are defined as outlier ones and indicated with a circle. All studies were approved by the Ethics Committee of the Tianjin Medical University, and informed consent was obtained from all patients.

**Tumor xenografts**. MCF-7 cells were plated and infected in vitro with lentiviruses carrying control shRNA, USP52 shRNA. Then, $3 \times 10^6$ viable MCF-7 cells in 200 μl PBS were injected into the mammary fat pads of 6- to 8-week-old athymic female mice (BALB/c; Charles River, Beijing, China). Six animals randomly assigned per group were used in each experiment. Sample size estimate was based on xenograft assays from literatures. 17-β-Estradiol (E2) pellets (0.72 mg per pellet, 60 day release; Innovative Research of America, Sarasota, FL) were implanted one day before the tumor cell injection. Tumors were measured weekly using a Vernier calliper and the volume was calculated according to the formula: $\pi/6 \times length \times width^2$. The measurement and data processing were done with blinding. All animals were killed at the end of the experiment and included into the analysis. The study was approved by the Animal Care Committee of Tianjin Medical University.

**MTS assay**. Cells were plated on 96-well dishes at a density of $5 \times 10^3$ cells/well. After overnight incubation, cells were treated with CPT or exposed to IR at the indicated dosages. Then, CellTiter 96® Aqueous One Solution Reagent (G3582, Promega) was added to each well according to the manufacturer's instructions. After 1 h incubation, cell viability was determined by measuring the absorbance at 490 nm using a 550 BioRad plate-reader (Bio-Rad, Hertfordshire, UK).

**X-ray irradiation**. IR was delivered by an X-ray generator (Radsource Corporation RS2000 PRO, 160 kV, 25 mA) according to the manufacturer's instructions.

**Statistical analysis**. Data from biological triplicate experiments are presented with error bar as mean ± S.D. Two-tailed unpaired Student's *t*-test was used for comparing two groups of data. Analysis of variance (ANOVA) with Bonferroni's correction was used to compare multiple groups of data. A *P*-value of less than 0.05 was considered significant. All of the statistical testing results were determined by SPSS 20.0 software. Before statistical analysis, variation within each group of data and the assumptions of the tests were checked.

**Study Approval**. All procedures involving animals were approved by the Ethics Committee of the Tianjin Medical University and followed the NIH Guide for the Care and Use of Laboratory Animals[65]. All studies associated with patient samples were approved by the Ethics Committee of the Tianjin Medical University, and informed consent was obtained from all patients.

**Data Availability**. All relevant data are available from the authors on request.

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

## Acknowledgements

This work was supported by grants (81722036 and 31671474 to L.S., 81502408 to L.L., and 81602511 to N.Y.) from the National Natural Science Foundation of China, grant (to L.S.) from Excellent Talent Project of Tianjin Medical University, and grant (IRT13085 to J.Y.) from Project for Innovative Research Team of Ministry of Education. We thank Pr. Baocun Sun and Pr. Xiulan Zhao (Tianjin Medical University, Tianjin, China) for providing help on IHC experiments.

## Author contributions

S.Y., L.L., and L.S. designed the research studies; S.Y., L.L., C.C., N.S., and Y.W. conducted experiments; S.Y., L.L., C.C., N.S., Y.W., S.M., Q.Z., X.D., N.Y., F.Y., S.T., K.Z., and T.S. acquired data; S.Y., L.L., C.C., N.S., J.Y., Z.Y., S.W., and L.S. analyzed data; S.Y., L.L., and L.S. wrote the manuscript.

## Additional information

**Competing interests:** The authors declare no competing interests.

