## [Peer Review File(PDF 627 kb) · Nature Communications]

Reviewer #1 (Remarks to the Author):

The study by Yang et al on the role of USP52 in deubiquitinating and stabilizing the histone chaperone ASF1A represents a large amount of impressive work.

Major points:

(i) the degree of in vitro activity appears to be low and this a concern. I don't think it is correct to state that the recombinant UCH domain "efficiently cleaves K48-linked chains" is correct. It is unclear to me (I could not find the information, maybe my mistake) on at least approximately how much enzyme and substrate that was used. The blots show cleavage of Ub4 to Ub3. The in vivo data suggest more effective activity (on longer chains, is the Ub4 a poor substrate?). I do not quite understand the discussion on the middle of page 10.

(ii) It would be nice to if the authors would have tested the effect on NEM or Ub-aldehyde on cleavage activity (especially considering the possible Cys-His juxtaposition mechanism).

(iii) Fig 4G is confusing. It does not look like the stability during the CHX chase is the same in the controls in the upper and lower panels (shorter in the lower panel, then stabilised by USP52 overexpression resulting in a half-life similar to the control in the upper panel). This does not look right.

(iv) For readers that are not in the field of chromatin assembly (like myself), the micrococcal nuclease experiments are not well explained (page 15). MNase cleaves between nucleosomes, I do not quite understand the logic of the experiment. Might very well be ignorance on my part, but the authors could explain this better.

(v) The association with breast cancer is in my opinion over-advertized. The association could be argued to be due to the rate of cell proliferation of any cell type or any cancer type. There is a large amount of very strong data presented, in my opinion this association could be downplayed a bit. That USP52 depleted MCF7 cells do not form tumors may (again) not be surprising since they do not proliferate that well.

Minor points:

(i) page 5: the number of active DUBs is below 100, closer to 80.

(ii) it is stated on page 6 that the catalytic domain lacks (classical) catalytic residues and in the Discussion that these residues may in fact be present, although somewhat displaced. Reading the manuscript I was confused by the apparent lack of catalytic residues and it would have helped if the information that they may actually be there would have been helpful.

(iii) there is so much information for the reader. Some of this could go into supplements, an obvious candidate being Fig 1C (considering Fig 1A, it would be fine to refer to a supplement that no other chains are cleaved to be GST-USP52).

The manuscript needs some language editing, there are sections that are unclear. One example is sentence #2 in the Abstract.

Reviewer #2 (Remarks to the Author):

The manuscript by Yang et al. investigated novel interacting partners of the histone chaperone ASF1A. They focus on USP52, a deubiquitinase proposed to act on ASF1A, and to stabilize its expression. They further add that USP52 by stabilizing the chaperone ASF1A could impact breast carcinogenesis.

While the topic concerning modifications of histone chaperones to regulate their function is of general interest, the previous report by Wang et al, 2015 already showing that the RAD6-MDM2 ubiquitin ligase machinery can regulate ASF1A degradation in human cells, limits the novelty of the present manuscript. In addition, the positive correlation between the high expression of ASF1A and H3K56Ac and tumorigenesis has also been reported previously for a series of tumors (Das et al, 2009). Thus, overall the main claims in this paper seem to add only an incremental advancement to the field.

Nevertheless, the authors provide insights into the molecular mechanism linking breast carcinogenesis via ASF1A stabilization that is mediated by USP52. Using techniques stable mammary tumor cell lines (MCF-7 cells) with stably integrated FLAG-ASF1A, by affinity purification combined with mass spectrometry they co-isolated a number of proteins with ASF1A. Among these proteins, they identified USP52 and demonstrated its ASF1A-deubiquitination activity through several assays. However, none of their assays enabled to identify specific residues of ASF1A that are targeted by USP52 neither in vitro, nor in vivo. Notably, for the latter, since the authors do not assess the ubiquitination of endogenous ASF1A, the issue remains unclear. Importantly, when assessing effect on chromatin status, the MNase digestion profiles provided do not reflect a clear importance of USP52-mediated ASF1A stabilization for global chromatin organization.

Finally, while the authors suggest that deubiquitination of the histone chaperone ASF1A by USP52 is involved in cancer progression, it is based on a limited number of experiments and additional

information is needed to strengthen their case, for example they could have analyzed the levels of ASF1A and USP52 in distinct breast tumour subtypes, and based on these information they could deepen their understanding. The addition of appropriate controls and improved presentation of the data are required to elevate the quality of the manuscript. The authors will find below the comments (in the order of the figures) that should be addressed to strengthen and improve the manuscript. We hope these comments can be helpful.

In summary, the manuscript suffers from novelty, the data supporting the main conclusions are still limited. Technically, a number of controls are missing, and these technical drawbacks severely affect the quality of the arguments. It seems thus premature to be considered for publication.

Comments:

1. Detailed characterization of ASF1A ubiquitination is required to add substantial novelty to the existing findings. Mass spectrometry data should be exploited further to identify potential ubiquitination sites on ASF1A. If ubiquitinated ASF1A peptides cannot be easily detected in the current dataset, it will be informative to perform the same analysis using USP52 depleted cells or in the presence of proteasome-inhibitors.
2. In Figure 1E, the authors show the physical association of ASF1A with USP52. The USP52 protein is not seen in inputs from the nuclear fraction. The authors should demonstrate whether USP52 is detectable in different cellular fractions by western blotting. Thus, it is not possible to determine whether USP52 is present in the nucleus, and if so, it can interact with ASF1A. Also, ASF1A levels should be included in each blot to demonstrate the efficiency of the IP experiment (also for Figure 1F).
3. In Figure 1D, the chromatographic elution fractions mentioned in the text for the peak of ASF1A-FLAG and USP52 co-purification (fractions 8 and 9) do not correspond to the fractions in the figure, rather the peak is observed in fraction 15. It is not clear what the labels A1, A3 etc. on the X axis represent and thus need clarification.
4. Based on the data presented in Figure 3 showing the deubiquitination of ASF1A mediated by USP52, it is unclear whether ASF1A is ubiquitinated at one or more residues. Importantly, the

authors only detect ubiquitination of tagged ASF1A throughout the manuscript, without assessing the ubiquitination of endogenous ASF1A.

5. Using HeLa cell extracts, the authors found that ASF1A is deubiquitinated by USP52 and they conclude that K129R-ASF1A is the predominate site “targeted” by USP52. However, in Figure 3G, it is not convincing that K129R-ASF1A is the major site that is “targeted” by USP52 for ASF1A deubiquitination. The mutant residues K41R-ASF1A and K129R-ASF1A show similar effects and could both be required for USP52-mediated deubiquitination. Moreover, it is not evident that these K residues are “targets” of deubiquitination or are merely required to recruit USP52 to other target sites on ASF1A.

6. The text states that six lysine residues in ASF1A were individually replaced with alanine (K129A), while Figure 3G depict K129R, K134R and other arginines. This needs clarification whether the residues were mutated to arginine or alanine.

7. In Figure 5A, it is hard to see the FACS profile and labeling of the X axis is unclear. This panel needs to be replaced with clearer profiles.

8. In Figure 5B, the MNase digestion profile after cell synchronization showed that overexpression of USP52 contributes to the promotion of chromatin assembly in S phase cells. The authors further claim that USP52-dependent ASF1A stabilization regulates chromatin assembly via H3K56Ac. The MNase digestion shows decreased nuclease sensitivity when FLAG- USP52 is overexpressed (gel 5 from the left), which is not comparable to the profile of Dox-induced ASF1A overexpression upon USP52 depletion (gel 2 from left). The MNase digestion profile following overexpression of ASF1A alone (maintaining wild-type USP52) is missing. This background would provide a context where ASF1 levels are increased thereby mimicking the stabilization of ASF1A by USP52. Thus, in the specific conditions tested, it is unclear that chromatin assembly is mediated by USP52 in an ASF1A dependent manner.

9. In Figure 5D, the authors claim that depletion of either USP52 or ASF1A led to an increase in the number of EdU positive cells (S phase). However, the fluorescence signal is very low in most panels making it difficult to visualize the increase in the number of EdU positive cells in the USP52 siRNA and ASF1A siRNA conditions (the EdU staining in these cells is hardly visible). The scale bars are also missing from the fluorescent images.

10. In Figure 6A, using immunohistochemical staining, the authors found that ASF1A and USP52 expression levels are significantly correlated. However, the images are too small to visualize the

staining. Similarly, in Figure 6F, increased magnification of the immunohistochemical images should be presented.

11. In Figure 7A, the relative cell viability in control siRNA, USP52 siRNA and ASF1A siRNA does not change drastically. The figure also lacks significance values. The authors do not comment on this in the text.

RE: MS# NCOMMS-17-22300

Title: USP52 Acts as a Deubiquitinase and Promotes Histone Chaperone ASF1A Stabilization

Response to Reviewer #1's comments-

The study by Yang et al on the role of USP52 in deubiquitinating and stabilizing the histone chaperone ASF1A represents a large amount of impressive work.

Major points and response:

Reviewer #1:

(i) the degree of in vitro activity appears to be low and this a concern. I don't think it is correct to state that the recombinant UCH domain "efficiently cleaves K48-linked chains" is correct. It is unclear to me (I could not find the information, maybe my mistake) on at least approximately how much enzyme and substrate that was used. The blots show cleavage of Ub4 to Ub3. The in vivo data suggest more effective activity (on longer chains, is the Ub4 a poor substrate?). I do not quite understand the discussion on the middle of page 10.

Authors: We thank the reviewer's constructive comments. We agree with the reviewer's point that it is not appropriate to state that the recombinant UCH domain "efficiently cleaves K48-linked chains". The word "efficiently" has been removed in the text of the revision. For *in vitro* deubiquitination assay, the detailed information about the amounts of enzyme and substrate that were used has been added to the corresponding Figure Legend of the revision. Specifically, 1 μg of ubiquitin linkages was incubated with 1, 2 or 3 μg of recombinant enzymes in experiments of previous submission. To address the reviewer's concern on the activity of UCH domain, we adjusted the molecular ratio of enzyme to substrate to examine the effect of NEM or cysteine mutations on the cleavage activity of USP52 UCH domain, and the results in the control group indicated that the recombinant UCH domain (3 μg) could efficiently hydrolyze K48- or K63-linked tetra-ubiquitin chains (0.5 μg) with the accumulation of shorter chains and free ubiquitin (Figure 2E and Figure 2F).

We also realized that the *in vitro* purified UCH domain of USP52 prefers to cleave shorter ubiquitin linkages. In particular, K48-linked di-ubiquitins could be more efficiently hydrolyzed (panel 1 of the third line in original Figure 2D, now shown as Figure 2C) than K48-linked tetra-ubiquitins (panel 3 of the second line in original Figure 2D, now shown as Figure 2C). Interestingly, it appears that endogenous or exogenous purified full length USP52 prefers to cleave longer ubiquitin chains conjugated on ASF1A and displays higher catalytic activity than the *in vitro* purified recombinant UCH

domain. We speculate that intra-protein collaborations, protein-protein interactions, post-translational modifications or a combination of these, could potentially enhance the enzymatic activity of USP52 or impact on its topology preference of poly-ubiquitin chains. Similar to our observations, recent studies reported that the deubiquitinase MINDY-1 catalytic domain is less efficient in cleaving long K48-polyubiquitin chains compared to the full-length deubiquitinase¹. Another pertinent example supporting the above deduction comes from the understanding of catalytic activity modulation of deubiquitinase USP7, whose C-terminal ubiquitin like (UBL) domain markedly promotes its deubiquitinating activity and this effect could be further allosterically activated by the metabolic enzyme GMP-synthetase (GMPS)². We have incorporated these interpretations into the Discussion on the last paragraph of page 27.

On the middle of page 10 in previous submission, we aimed to explain why Sf9 cells purified UCH domain of USP52 exhibited catalytic activity, while prokaryotic cells purified protein failed to hydrolyze ubiquitin chains. Also, we want to discuss why UCH domain of USP52 cleaves ubiquitin chains in a relative slow kinetics compared with certain canonical deubiquitinases, which usually starts cleaving ubiquitin linkages after minutes of reaction and almost fully hydrolyzes the isopeptide bond in less than one hour^{3, 4}. Given that specific protein folding features or post-translational modification could impact on higher structure conformation and potentially coordinate or rearrange the catalytic center of enzymes⁵, we deduced that the UCH domain of USP52 purified from Sf9 cells likely undergoes conformational changes to gradually accommodate with ubiquitin substrates under the help of specific post-translational modifications or folding features thus eventually elicits detectable enzymatic activity, albeit with a slower kinetics. To address the reviewer's concern, we have reworded this part as indicated in the revision.

Reviewer #1:

(ii) It would be nice to if the authors would have tested the effect on NEM or Ub-aldehyde on cleavage activity (especially considering the possible Cys-His juxtaposition mechanism).

Authors: We have examined the effect of NEM on the cleavage activity of USP52 UCH domain, and the results demonstrated that the addition of NEM almost completely abolished the catalytic activity of USP52 against K48 or K63 linked tetra-ubiquitins (Figure 2E). In this experiment, USP7 was taken as a positive control (Figure 2E). Similarly, inclusion of NEM in the reactions also completely abolished such activity of USP52 against polyubiquitinated ASF1A (Figure S3C). These results suggest the involvement of one or more cysteine residues in catalysis.

Reviewer #1:

(iii) Fig 4G is confusing. It does not look like the stability during the CHX chase is the same in the controls in the upper and lower panels (shorter in the lower panel, then stabilised by USP52 overexpression resulting in a half-life similar to the control in the upper panel). This does not look right.

Authors: We thank the reviewer's comment and apologize for the confusion. In the original blots of the lower panel of Figure 4G, lighter exposure blots were used to highlight the effect of USP52 overexpression in extending the half-life of ASF1A. To avoid confusion, we have provided longer exposure blots to replace the original one in the lower panel of Figure 4G. Additional experiments corresponding to original Figure 4G were also performed (Figure S4A and S4B). The quantitation and statistical analysis of these results from biological triplicate experiments have been provided in Figure S4A and S4B.

Reviewer #1:

(iv) For readers that are not in the field of chromatin assembly (like myself), the micrococcal nuclease experiments are not well explained (page 15). MNase cleaves between nucleosomes, I do not quite understand the logic of the experiment. Might very well be ignorance on my part, but the authors could explain this better.

Authors: We apologize for the unclear description of this experiment. Micrococcal nuclease (MNase) preferentially cleaves internucleosomal (linker) DNA or nucleosome free DNA and reveals the size and regularity of assembled chromatin by producing a ladder of 146-200 bp DNA molecules, the typical size of one nucleosome unit⁶. Generally, chromatins with high density of nucleosome occupancy are reluctant to be digested in the presence of low concentration of MNase. Thereby, MNase experiments are routinely used to monitor chromatin assembly or examine chromatin accessibility^{7, 8}. Chromatin assembly defects are commonly manifested by a reduction or an impairment of nucleosome occupancy, the state of which will result in chromatins more sensitive to MNase cleavage. In a previous study, MNase digestion assay was used to examine the effect of ASF1A on chromatin assembly⁶. In our study, to understand whether USP52-promoted ASF1A stabilization plays a role in replication-coupled chromatin assembly, cells enriched in S phase were harvested and newly replicated chromatins were isolated for the MNase digestion assay. To address the reviewer's concern, we have re-described the rationale and principle of this experiment in the revision.

Reviewer #1:

(v) The association with breast cancer is in my opinion over-advertized. The association could be argued to be due to the rate of cell proliferation of any cell type or any cancer type. There is a large amount of very strong data presented, in my opinion this association could be downplayed a bit. That USP52 depleted MCF7 cells do not form

tumors may (again) not be surprising since they do not proliferate that well.

Authors: We agree with the reviewer's point that the association of USP52-promoted ASF1A stabilization with breast cancer is over-advertized. In the revision, we have softened this argument and retitled our manuscript as "USP52 Acts as a Deubiquitinase and Promotes Histone Chaperone ASF1A Stabilization". Also, we replaced our statement that "USP52-regulated ASF1A stabilization promotes breast carcinogenesis" with "USP52-promoted ASF1A stabilization is implicated in breast carcinogenesis" in the text. According to the suggestion from the Reviewer#2, we enlarged the sample sizes of invasive ductal breast cancer and analyzed the expression levels of ASF1A and USP52 in other histological types of breast tumor. The results are provided in Figure 6A to replace the previous data.

Minor points and response:

Reviewer #1:

(i) page 5: the number of active DUBs is below 100, closer to 80.

Authors: We thank the reviewer's reminding and the number has been changed in the text.

Reviewer #1:

(ii) it is stated on page 6 that the catalytic domain lacks (classical) catalytic residues and in the Discussion that these residues may in fact be present, although somewhat displaced. Reading the manuscript I was confused by the apparent lack of catalytic residues and it would have helped if the information that they may actually be there would have been helpful.

Authors: We apologize for the unclear description. Sequence alignment indicates that the conserved cysteine residue and histidine residue in the triad are replaced by alanine (A526) and serine (S867), respectively, in the UCH domain of USP52 (Figure S2A). We hypothesized that C528 or C530 (adjacent to A526) and histidine adjacent to S867 (like H871) could be rearranged to form newly active center thus enable the UCH domain of USP52 to hydrolyze ubiquitin chains.

Although NEM treatment almost completely abolished the catalytic activity of USP52 against tetra-ubiquitins (Figure 2E) or poly-ubiquitinated ASF1A (Figure S3C), mutational analysis showed that C528 and C530 do not appear to be critical for the catalytic activity of USP52 (Figure 2F). In the future, functional analysis of distal or other proximal cysteine/histidine residue mutants with deubiquitination assays, and crystal structure analysis of the UCH domain of USP52 purified from Sf9 cells together with

different types of ubiquitin linkages, will be helpful in characterizing the essential residues required for catalytic activity and understanding the molecular mechanism of USP52 hydrolyzing the isopeptide bond of distinct ubiquitin linkages. The experiments have been described in the last paragraph on page 11 in the text and the discussion on these experiments has been incorporated into the last paragraph on page 28 of the revision.

Reviewer #1:

(iii) there is so much information for the reader. Some of this could go into supplements, an obvious candidate being Fig 1C (considering Fig 1A, it would be fine to refer to a supplement that no other chains are cleaved to be GST-USP52.

Authors: We thank the reviewer's suggestion. Original Figure 1C, Figure 1H, Figure 1I, Figure 2C, Figure 3D, Figure 3E, Figure 5A and Figure 6E have been moved to Supplemental Figures, and original Figure 3H have been replaced with new data and shown as Figure S3E.

Reviewer #1:

(iv) The manuscript needs some language editing, there are sections that are unclear. One example is sentence #2 in the Abstract.

Authors: The manuscript has been carefully edited.

RE: MS# NCOMMS-17-22300

Title: USP52 Acts as a Deubiquitinase and Promotes Histone Chaperone ASF1A Stabilization

Response to Reviewer #2's comments-

Reviewer #2:

The manuscript by Yang et al. investigated novel interacting partners of the histone chaperone ASF1A. They focus on USP52, a deubiquitinase proposed to act on ASF1A, and to stabilize its expression. They further add that USP52 by stabilizing the chaperone ASF1A could impact breast carcinogenesis.

While the topic concerning modifications of histone chaperones to regulate their function is of general interest, the previous report by Wang et al, 2015 already showing that the RAD6-MDM2 ubiquitin ligase machinery can regulate ASF1A degradation in human cells, limits the novelty of the present manuscript. In addition, the positive correlation between the high expression of ASF1A and H3K56Ac and tumorigenesis has also been reported previously for a series of tumors (Das et al, 2009). Thus, overall the main claims in this paper seem to add only an incremental advancement to the field.

Nevertheless, the authors provide insights into the molecular mechanism linking breast carcinogenesis via ASF1A stabilization that is mediated by USP52. Using techniques stable mammary tumor cell lines (MCF-7 cells) with stably integrated FLAG-ASF1A, by affinity purification combined with mass spectrometry they co-isolated a number of proteins with ASF1A. Among these proteins, they identified USP52 and demonstrated its ASF1A-deubiquitination activity through several assays. However, none of their assays enabled to identify specific residues of ASF1A that are targeted by USP52 neither in vitro, nor in vivo. Notably, for the latter, since the authors do not assess the ubiquitination of endogenous ASF1A, the issue remains unclear. Importantly, when assessing effect on chromatin status, the MNase digestion profiles provided do not reflect a clear importance of USP52-mediated ASF1A stabilization for global chromatin organization.

Finally, while the authors suggest that deubiquitination of the histone chaperone ASF1A by USP52 is involved in cancer progression, it is based on a limited number of experiments and additional information is needed to strengthen their case, for example they could have analyzed the levels of ASF1A and USP52 in distinct breast tumour subtypes, and based on these information they could deepen their understanding. The addition of appropriate controls and improved presentation of the data are required to elevate the quality of the manuscript. The authors will find below the comments (in the order of the figures) that should be addressed to strengthen and improve the manuscript. We hope these comments can be helpful.

In summary, the manuscript suffers from novelty, the data supporting the main conclusions are still limited. Technically, a number of controls are missing, and these technical drawbacks severely affect the quality of the arguments. It seems thus premature to be considered for publication.

Authors: We appreciate the reviewer's constructive criticisms. To comply with the suggestion of the Reviewer #1 (major point 5), we have retitled our manuscript as "USP52 Acts as a Deubiquitinase and Promotes Histone Chaperone ASF1A Stabilization".

We agree with the reviewer that the previous reports by Wang et al., 2015⁹ and Das et al., 2009¹⁰, to certain extent, limits the novelty of our findings. However, the main novelty of our work is based on the weight of the following observations: 1) Ubiquitin-specific peptidase 52 (USP52), a member of the ubiquitin-specific protease superfamily, has been considered as a pseudo-deubiquitinase according to crystal structure analysis of USP52 from *Neurospora crassa* or *Saccharomyces cerevisiae*^{11, 12}. Yet, whether USP52 is capable of hydrolyzing ubiquitin linkages remains an open question. 2) Although histone chaperone ASF1A has been reported to be dysregulated in multiple tumors, the underlying molecular mechanism that how the abundance and function of ASF1A are regulated remains less understood and specific deubiquitinase for ASF1A has not been identified. Through a body of work, we discovered that USP52 is able to remove ubiquitins either from specific types of poly-conjugated ubiquitin chains or K48-linked polyubiquitinated ASF1A. Second, we revealed that USP52 stabilizes ASF1A through counteracting with ASF1A ubiquitination. To our knowledge, USP52 is likely the first deubiquitinase identified to date for ASF1A deubiquitination. Our findings provide valuable insight on the catalytic activity of USP52 and advance the understanding of the maintenance of ASF1A abundance thus the level of H3K56Ac. Third, we showed that USP52 promotes chromatin assembly through stabilizing ASF1A, and revealed a potential role of USP52 in breast carcinogenesis and cellular resistance to DNA damage in breast cancer cells.

According to the reviewer's suggestion, we have tried to identify specific residues of ASF1A that are targeted by USP52 through mass spectrometry analysis (Figure 3H and Supplemental Table 2), *in vitro* and *in vivo* deubiquitination assays (Figure 3G, Figure S3D and Figure S3F), and CHX chase assays (Figure S4C). These results revealed that K129 is the predominant targeting site of USP52 for controlling ASF1A deubiquitination and stabilization. Also, we demonstrated that USP52 overexpression is associated with a decrease in the level of ubiquitin species conjugated on endogenous ASF1A (Figure 3D and Figure 3E).

To reflect an importance of USP52 in global chromatin organization and identify the role

of USP52-mediated ASF1A stabilization in this process, we examined the effect of ASF1A or USP52 overexpression on replication-coupled chromatin assembly (Figure S5B), and simultaneously, we repeated experiments in gel 2 and gel 5 of previous Figure 5B (now shown as Figure 5A). We demonstrated that forced expression of USP52 alone mimics the effect of ASF1A overexpression (Figure S5B) and the MNase digestion profiles of ASF1A or USP52 overexpression in each panel are largely consistent (Figure 5A, panel 2 and panel 5; Figure S5B). The argument that USP52 promotes chromatin assembly through stabilizing ASF1A could be supported by these results and the following observations: 1) either USP52 or ASF1A deficiency led to increased chromatin sensitivity to MNase digestion (previous Figure 5B, panel 1; now shown as Figure 5A, panel 1); 2) the defects of replication-coupled nucleosome assembly associated with USP52 depletion could be ameliorated by ASF1A overexpression (previous Figure 5B, panel 2; now shown as Figure 5A, panel 2); 3) USP52 gain of function resulted in chromatin resistance to MNase digestion, the effects of which could be reverted by ASF1A depletion (previous Figure 5B, panel 3; now shown as Figure 5A, panel 3); and 4) synergistically depletion of USP52 and ASF1A had no additive effect (previous Figure 5B, panel 4; now shown as Figure 5A, panel 4).

We have analyzed the levels of ASF1A and USP52 in other subtypes of breast tumor, including the invasive lobular and intraductal breast cancer. Also, we enlarge the sample sizes of invasive ductal breast cancer and tumor adjacent samples that have been analyzed previously. IHC examination indicates the expression levels of ASF1A and USP52 are elevated in different histologic types of breast carcinoma, albeit to variable extent, and the expression level of ASF1A positively correlates with that of USP52 in breast cancer. New data has been provided in Figure 6A to replace the previous one.

About the control experiments, we had examined the interaction of USP52 with PAN3 (Figure 1B) or ASF1B as negative control (previous Figure 1C, now shown as Figure S1B), and we also used USP7 as a negative control when examined the effect of USP52 depletion on the expression of ASF1A (Figure 2C) in the previous submission. Nevertheless, we agree with the reviewer's point that some experiments still need appropriate controls. In the revision, we examined the effect of alkylating reagent N-ethylmaleimide (NEM) on the catalytic activity of USP52 with *in vitro* deubiquitination assay, in which USP7 was taken as a positive control (Figure 2E). In the MNase digestion assay, we include ASF1A as a positive control to examine the effect of USP52 on global chromatin organization (Figure S5B). Finally, we have tried to improve the presentation of our data from experimental performance to language editing. Hopefully, these additional experiments and revised presentation could address the reviewer's concern.

Comments and response:

Reviewer #2:

1. Detailed characterization of ASF1A ubiquitination is required to add substantial novelty to the existing findings. Mass spectrometry data should be exploited further to identify potential ubiquitination sites on ASF1A. If ubiquitinated ASF1A peptides cannot be easily detected in the current dataset, it will be informative to perform the same analysis using USP52 depleted cells or in the presence of proteasome-inhibitors.

Authors: We thank the reviewer's comment and suggestion. To characterize ASF1A ubiquitination sites, we purified FLAG tagged ASF1A from USP52 depleted HeLa cells. Mass spectrometry analysis revealed K129, but not other lysine residues, carries di-Glycine remnant after trypsin digestion (Figure 3H and Supplemental Table 2).

Reviewer #2:

2. In Figure 1E, the authors show the physical association of ASF1A with USP52. The USP52 protein is not seen in inputs from the nuclear fraction. The authors should demonstrate whether USP52 is detectable in different cellular fractions by western blotting. Thus, it is not possible to determine whether USP52 is present in the nucleus, and if so, it can interact with ASF1A. Also, ASF1A levels should be included in each blot to demonstrate the efficiency of the IP experiment (also for Figure 1F).

Authors: To address the reviewer's concern, we re-performed the experiments in previous Figure 1E. Cellular component fractionation and Western blotting analysis indicate that USP52 is mainly collected from cytosol and nearly absent in nucleus (Figure 1D, right panel). Interestingly, ASF1A also exhibits evidently cytoplasmic distribution, although the abundance is not as much as that in the nuclear fraction (Figure 1D, right panel). Moreover, immunostaining followed by confocal microscopy analysis confirmed USP52 mainly co-localizes with ASF1A in cytoplasm (Figure 1E). Thereby, we conceived that USP52 interacts with and stabilizes ASF1A predominantly in cytoplasm. According to the reviewer's suggestion, we have repeated the co-immunoprecipitation experiments in original Figure 1F and 1G. New data has been provided in Figure 1F and Figure 1G to replace the previous ones.

Reviewer #2:

3. In Figure 1D, the chromatographic elution fractions mentioned in the text for the peak of ASF1A-FLAG and USP52 co-purification (fractions 8 and 9) do not correspond to the fractions in the figure, rather the peak is observed in fraction 15. It is not clear what the labels A1, A3 etc. on the X axis represent and thus need clarification.

Authors: The largest peak (B4, fraction number 21) in the chromatographic elution profile likely represents the excess $3 \times$ FLAG peptides that were used to competitively eluate FLAG-ASF1A containing protein complex from FLAG agarose gel. The label A1

on the X axis represents elution starting point thus fraction number 1. The column was eluted at a flow rate of 0.5 ml/min and fractions were collected every two minutes. The numbers below X axis in the original data is the accumulation volume of elution buffer that were used. To avoid confusion, we have replaced these labels with actual fraction numbers. We also re-examined the elution positions of calibration proteins with known molecular masses and adjusted the arrows representing molecular weights to the appropriate fractions as indicated. We thank the reviewer for pointing this and apologize for the confusion we had brought.

Reviewer #2:

4. Based on the data presented in Figure 3 showing the deubiquitination of ASF1A mediated by USP52, it is unclear whether ASF1A is ubiquitinated at one or more residues. Importantly, the authors only detect ubiquitination of tagged ASF1A throughout the manuscript, without assessing the ubiquitination of endogenous ASF1A.

Authors: Deubiquitination assays with ASF1A KR mutants suggested that ASF1A could be ubiquitinated at more than one residue, as none of these KR mutations are able to eliminate ASF1A ubiquitination (Figure 3G and Figure S3D). Furthermore, mass spectrometry analysis, *in vitro* deubiquitination and CHX chase assays revealed that K129 is the predominant targeting site of USP52 for controlling ASF1A deubiquitination and stabilization. Specifically, we found trypsin digestion retrieved multiple peptides containing lysine residues, but K129 is the only lysine residue carrying di-Glycine remnant (Figure 3H and Supplemental Table 2), and *in vitro* deubiquitination assays and CHX chase assays with ASF1A/wt, ASF1A/K41R and ASF1A/K129R confirmed that USP52 stabilizes ASF1A through cleaving ASF1A/K129-linked polyubiquitin chains (Figure S3F and Figure S4C).

Next, we assessed the ubiquitination of endogenous ASF1A, the results showed that endogenous ASF1A is a ubiquitinated protein and USP52 overexpression resulted in a decrease in the level of ubiquitinated ASF1A species (Figure 3D). In addition, IP of cellular lysates with antibody against endogenous ASF1A followed by IB with anti-HA revealed that the levels of K48-linked ubiquitinated ASF1A species markedly decreased in USP52 overexpression cells (Figure 3E).

Reviewer #2:

5. Using HeLa cell extracts, the authors found that ASF1A is deubiquitinated by USP52 and they conclude that K129R-ASF1A is the predominate site “targeted” by USP52. However, in Figure 3G, it is not convincing that K129R-ASF1A is the major site that is “targeted” by USP52 for ASF1A deubiquitination. The mutant residues K41R-ASF1A and K129R-ASF1A show similar effects and could both be required for USP52-mediated deubiquitination. Moreover, it is not evident that these K residues are “targets” of

deubiquitination or are merely required to recruit USP52 to other target sites on ASF1A.

Authors: We appreciate the reviewer's comment. In the revision, we have replaced the previous IP blot corresponding to ASF1A/K41R deubiquitination with lighter exposure one, from which we can see that ubiquitin linkages conjugated on ASF1A/K41R could be removed by USP52 (Figure 3G). In addition, we repeated the deubiquitination assays and the results provided in Figure S3D confirmed that K129R-ASF1A, but not other ASF1A KR mutants, is resistant to ubiquitin cleavage by USP52.

Furthermore, *in vitro* deubiquitination assays with USP52 and HA-Ub/K48-only-conjugated ASF1A/wt, ASF1A/K41R or ASF1A/K129R purified from mammalian cells revealed that USP52 was able to remove the ubiquitin linkages conjugated on ASF1A/wt and ASF1A/K41R, but not that of ASF1A/K129R (Figure S3F). Meanwhile, CHX chase assays in MCF-7 cells stably expressing FLAG-ASF1A/K41R or FLAG-ASF1A/K129R showed that depletion of USP52 was associated with a decrease in the half-life of ASF1A/K41R, but not that of ASF1A/K129R (Figure S4C). Importantly, mass spectrometry analysis identified K129 as the only lysine residue carrying di-Glycine remnant (Figure 3H). Although other ubiquitination sites on ASF1A possibly exist, the above results support the notion that K129 is the major site targeted by USP52 for ASF1A deubiquitination and stabilization.

To rule out the possibility that these lysine residues are not “targets” of deubiquitination, but merely required to recruit USP52 to the other target sites on ASF1A, we then examined the association of each ASF1A KR mutant with USP52. The co-immunoprecipitated protein complex was loaded on the same gel and Western blotting analysis revealed that similar amount of USP52 could be co-immunoprecipitated with almost equal abundance of ASF1A KR mutants and wild type ASF1A (Figure S3E), indicating that these lysine residues are not involved in the interaction between ASF1A and USP52.

Reviewer #2:

6. The text states that six lysine residues in ASF1A were individually replaced with alanine (K129A), while Figure 3G depict K129R, K134R and other arginines. This needs clarification whether the residues were mutated to arginine or alanine.

Authors: We apologized for this confusion. We have confirmed that lysine residues were mutated to arginine but not alanine and the mistake has been corrected.

Reviewer #2:

7. In Figure 5A, it is hard to see the FACS profile and labeling of the X axis is unclear. This panel needs to be replaced with clearer profiles.

Authors: The FACS profile in original Figure 5A has been replaced with clearer images and labels, and the results have been moved to Figure S5A.

Reviewer #2:

8. *In Figure 5B, the MNase digestion profile after cell synchronization showed that overexpression of USP52 contributes to the promotion of chromatin assembly in S phase cells. The authors further claim that USP52-dependent ASF1A stabilization regulates chromatin assembly via H3K56Ac. The MNase digestion shows decreased nuclease sensitivity when FLAG- USP52 is overexpressed (gel 5 from the left), which is not comparable to the profile of Dox-induced ASF1A overexpression upon USP52 depletion (gel 2 from left). The MNase digestion profile following overexpression of ASF1A alone (maintaining wild-type USP52) is missing. This background would provide a context where ASF1 levels are increased thereby mimicking the stabilization of ASF1A by USP52. Thus, in the specific conditions tested, it is unclear that chromatin assembly is mediated by USP52 in an ASF1A dependent manner.*

Authors: We appreciate the reviewer's comments. Indeed, the MNase digestion profile of USP52 overexpression (lane 3 of gel 5 in previous Figure 5B) appears to be incompatible with that in lane 4 of gel 2, lane 3 of gel 3 and lane 3 of gel 6, where ASF1A or USP52 is forced expression. We assume this discrepancy likely comes from inappropriate experimental conduction. For example, nuclei aggregation could prohibit MNase activity thus led to accumulation of longer chromatin fibers after digestion. To reflect a clear importance of USP52-mediated ASF1A stabilization for global chromatin organization, we repeated experiments in gel 2 and gel 5 in previous Figure 5B. Simultaneously, we also, in parallel, examined the effect of ASF1A or USP52 overexpression on replication-coupled chromatin assembly. We demonstrated that the MNase digestion profiles of ASF1A or USP52 overexpression in each panel are largely consistent (lane 4 of gel 2, lane 3 of gel 3, lane 3 of gel 5 and lane 3 of gel 6 as shown in Figure 5A of the revision) and the effect of USP52 overexpression on replication-coupled chromatin assembly is comparable to that of ASF1A overexpression (Figure S5B). Together, these results support the argument that USP52 promotes chromatin assembly through stabilizing ASF1A.

Reviewer #2:

9. *In Figure 5D, the authors claim that depletion of either USP52 or ASF1A led to an increase in the number of EdU positive cells (S phase). However, the fluorescence signal is very low in most panels making it difficult to visualize the increase in the number of EdU positive cells in the USP52 siRNA and ASF1A siRNA conditions (the EdU staining in these cells is hardly visible). The scale bars are also missing from the fluorescent images.*

Authors: To visualize the EdU signals, the images have been converted into black and white thus EdU positive cells could be easily recognized and clearly counted, although the stainings in knockdown group are still weak. The results have been provided in Figure S5D. In addition, the scale bars have been added.

Reviewer #2:

10. In Figure 6A, using immunohistochemical staining, the authors found that ASF1A and USP52 expression levels are significantly correlated. However, the images are too small to visualize the staining. Similarly, in Figure 6F, increased magnification of the immunohistochemical images should be presented.

Authors: Increased magnification of the immunohistochemical images in these figures have been provided in the revision.

Reviewer #2:

11. In Figure 7A, the relative cell viability in control siRNA, USP52 siRNA and ASF1A siRNA does not change drastically. The figure also lacks significance values. The authors do not comment on this in the text.

Authors: We have repeated these experiments in biological triplicates, and the results and statistical analysis are provided in Figure 7A to replace the previous data. Cell viability examination indicated cells with USP52 or ASF1A depletion, albeit not drastically, were more sensitive to higher dose of IR treatment (Figure 7A).

References

1. Abdul Rehman SA, Kristariyanto YA, Choi SY, Nkosi PJ, Weidlich S, Labib K, Hofmann K, Kulathu Y. MINDY-1 Is a Member of an Evolutionarily Conserved and Structurally Distinct New Family of Deubiquitinating Enzymes. *Mol Cell* **63**, 146-155 (2016).
2. Faesen AC, Dirac AM, Shanmugham A, Ovaa H, Perrakis A, Sixma TK. Mechanism of USP7/HAUSP activation by its C-terminal ubiquitin-like domain and allosteric regulation by GMP-synthetase. *Mol Cell* **44**, 147-159 (2011).
3. Mevissen TE, Hospenthal MK, Geurink PP, Elliott PR, Akutsu M, Arnaudo N, Ekkebus R, Kulathu Y, Wauer T, El Oualid F, Freund SM, Ovaa H, Komander D. OTU deubiquitinases reveal mechanisms of linkage specificity and enable ubiquitin chain restriction analysis. *Cell* **154**, 169-184 (2013).
4. Meulmeester E, Kunze M, Hsiao HH, Urlaub H, Melchior F. Mechanism and consequences for paralog-specific sumoylation of ubiquitin-specific protease 25. *Mol Cell* **30**, 610-619 (2008).
5. Schultz DC, Ayyanathan K, Negorev D, Maul GG, Rauscher FJ, 3rd. SETDB1: a novel KAP-1-associated histone H3, lysine 9-specific methyltransferase that contributes to HP1-mediated silencing of euchromatic genes by KRAB zinc-finger proteins. *Genes Dev* **16**, 919-932 (2002).
6. Sanematsu F, Takami Y, Barman HK, Fukagawa T, Ono T, Shibahara K, Nakayama T. Asf1 is required for viability and chromatin assembly during DNA replication in vertebrate cells. *J Biol Chem* **281**, 13817-13827 (2006).
7. Li L, Shi L, Yang S, Yan R, Zhang D, Yang J, He L, Li W, Yi X, Sun L, Liang J, Cheng Z, Shi L, Shang Y, Yu W. SIRT7 is a histone desuccinylase that functionally links to chromatin compaction and genome stability. *Nat Commun* **7**, 12235 (2016).
8. Mieczkowski J, Cook A, Bowman SK, Mueller B, Alver BH, Kundu S, Deaton AM, Urban JA, Larschan E, Park PJ, Kingston RE, Tolstorukov MY. MNase titration reveals differences between nucleosome occupancy and chromatin accessibility. *Nat Commun* **7**, 11485 (2016).
9. Wang C, Chang JF, Yan H, Wang DL, Liu Y, Jing Y, Zhang M, Men YL, Lu D, Yang XM, Chen S, Sun FL. A conserved RAD6-MDM2 ubiquitin ligase

- machinery targets histone chaperone ASF1A in tumorigenesis. *Oncotarget* **6**, 29599-29613 (2015).
10. Das C, Lucia MS, Hansen KC, Tyler JK. CBP/p300-mediated acetylation of histone H3 on lysine 56. *Nature* **459**, 113-117 (2009).
 11. Schafer IB, Rode M, Bonneau F, Schussler S, Conti E. The structure of the Pan2-Pan3 core complex reveals cross-talk between deadenylase and pseudokinase. *Nat Struct Mol Biol* **21**, 591-598 (2014).
 12. Jonas S, Christie M, Peter D, Bhandari D, Loh B, Huntzinger E, Weichenrieder O, Izaurralde E. An asymmetric PAN3 dimer recruits a single PAN2 exonuclease to mediate mRNA deadenylation and decay. *Nat Struct Mol Biol* **21**, 599-608 (2014).

Reviewer #1 (Remarks to the Author):

The authors have responded to the points I raised and have improved the manuscript. A lot of work of work is presented. It would have been useful if a mutational analysis had been performed to identify the catalytic cysteine but I dont think it is reasonable to demand more data.

Reviewer #2 (Remarks to the Author):

The authors have successfully addressed all the concerns raised. The quality of the manuscript has substantially improved. Therefore, it is recommended as acceptable for publication.

RE: MS# NCOMMS-17-22300B

Title: USP52 Acts as a Deubiquitinase and Promotes Histone Chaperone ASF1A Stabilization

Response to Reviewer #1's comments-

The authors have responded to the points I raised and have improved the manuscript. A lot of work of work is presented. It would have been useful if a mutational analysis had been performed to identify the catalytic cysteine but I dont think it is reasonable to demand more data.

Authors: We thank the reviewer's comments. Indeed, mutational analysis had been performed to identify the catalytic cysteine of USP52 in our previous submission. However, *in vitro* deubiquitination assays revealed that C528A/C530A mutant is still active in hydrolyzing K48- or K63-linked tetra-ubiquitins, albeit with an evidently lower efficiency (Fig. 2f). These results indicate that one or both of the two cysteine residues is important but not essential for the enzymatic activity of USP52.

We agree with the reviewer that further mutational analysis with deubiquitination assays, as well as crystal structure analysis of the UCH domain of USP52 purified from Sf9 cells together with different types of ubiquitin linkages will be helpful in characterizing the essential residues required for catalytic activity and understanding the molecular mechanism of USP52 hydrolyzing the isopeptide bond of distinct ubiquitin linkages. This information has been incorporated into the DISCUSSION section on page 25 of the revised manuscript.